

# Retrieval of ash properties from IASI measurements

Lucy J. Ventress[1], Don Grainger[2], Gregory McGarragh[3], Elisa Carboni[2], and Andrew J. Smith[1]

[1]National Centre for Earth Observation, Atmospheric, Oceanic and Planetary Physics, University of Oxford, Parks Road, Oxford OX1 3PU, U.K.
[2]COMET, Atmospheric, Oceanic and Planetary Physics, University of Oxford, Parks Road, Oxford, OX1 3PU, U.K.
[3]Atmospheric, Oceanic and Planetary Physics, University of Oxford, Parks Road, Oxford, OX1 3PU, U.K.

*Correspondence to:* L. J. Ventress (lucy.ventress@physics.ox.ac.uk)

**Abstract.** A new optimal estimation algorithm for the retrieval of volcanic ash properties has been developed for use with hyperspectral satellite instruments such as the Infrared Atmospheric Sounding Interferometer (IASI). The retrieval method uses the wavenumber range $680$–$1200 \, \mathrm{cm}^{-1}$, which contains window channels, the $CO_2 \, \nu_2$ band (used for the height retrieval), and the $O_3 \, \nu_3$ band.

Assuming a single infinitely (geometrically) thin ash plume and combining this with the output from the radiative transfer model RTTOV, the retrieval algorithm produces the most probable values for the ash optical depth (AOD), particle effective radius, plume top height and surface temperature. A comprehensive uncertainty budget is obtained for each pixel. Improvements to the algorithm through the use of different measurement error covariance matrices is explored, comparing the results from a sensitivity study of the retrieval process using covariance matrices trained on either clear-sky or cloudy scenes. The result

exhibited that, due to the smaller variance contained within it, the clear-sky covariance matrix is preferable. However, if the retrieval fails to pass the quality control tests, the cloudy covariance matrix is implemented.

    The retrieval algorithm is applied to scenes from the Eyjafjallajökull eruption in 2010 and the retrieved parameters are compared to ancillary data sources. The ash optical depth gives an RMS difference of $0.46$ when compared to retrievals from the MODIS instrument for all pixels and an improved RMS of $0.2$ for low optical depths. Measurements from the FAAM and

DLR flight campaigns are used to verify the retrieved particle effective radius, with the retrieved distribution of sizes for the scene showing excellent consistency. Further, the plume top altitudes are compared to derived cloud-top altitudes from the CALIOP instrument and show agreement with RMS values of less than $1 \, \mathrm{km}$.

## 1   Introduction

The detection of volcanic ash and the retrieval of its properties has become a topic of increasing interest following the eruption

of Eyjafjallajökull in 2010. Volcanoes are responsible for the emission of large quantities of aerosol particles and gases, such as $H_2O$, $CO_2$ and $SO_2$, into the atmosphere. The particles created during a volcanic event are classified according to size with the smaller solid particles (radii $< 2 \, \mathrm{mm}$) referred to as volcanic ash (Schmid, 1981). These particles can have significant effects upon the Earth's radiation balance, air quality and the aviation industry; the worst outcome in the latter case resulting in engine failure (Grainger et al., 2013; Casadevall, 1994). Through the analysis of spectral information from satellite





infrared spectrometers (such as the Atmospheric Infrared Sounder, AIRS, the Tropospheric Emission Spectrometer, TES, and the Infrared Atmospheric Sounding Interferometer, IASI), the optical and physical properties of volcanic ash can be derived (e.g. the mass of ash contained within the plume) and these can be used to calculate the parameters most useful in ensuring safe air travel (Dubuisson et al., 2014). Several different approaches have been applied to the infrared spectra of different volcanic
plumes, including methods based upon optimal estimation (Clarisse et al., 2010; Francis et al., 2012; Pavolonis et al., 2013), singular value decomposition (Klüser et al., 2013) and split-window (Wen and Rose, 1994; Prata and Grant, 2001).

Presented here is a new optimal estimation algorithm for the retrieval of volcanic ash properties that has been developed for use with hyperspectral satellite instruments such as IASI. The retrieval method uses the wavenumber range $680-1200\,\mathrm{cm}^{-1}$, which contains window channels, the $CO_2$ $\nu_2$ band, and the $O_3$ $\nu_3$ band.

This paper takes the Oxford-RAL Retrieval of Aerosol and Cloud (ORAC) algorithm (Thomas et al., 2009a; Poulsen et al., 2012), which was successfully applied to the retrieval of volcanic $SO_2$ by Carboni et al. (2012) through the addition of a generalised error covariance matrix, and adapts it for use with volcanic ash. The method uses an optimal estimation retrieval algorithm to obtain probable values for the ash optical depth (AOD), particle effective radius, plume top height and surface temperature. The reliability of the retrieved parameters is discussed with a focus upon the validation of the height product, which,
in other methods is usually assumed to be some fixed value. Identifying the ash plume top height is a challenge for remote sensing as it is a critical parameter for the initialisation of algorithms that numerically model the evolution and transport of a volcanic plume (Grainger et al., 2013). Validation of the parameters is carried out through comparisons to the derived plume top height from the Cloud- Aerosol Lidar with Orthogonal Polarization (CALIOP), a retrieved AOD from the MODerate-resolution Imaging Spectroradiometer (MODIS) and particle effective radius measurements from the Facility for Airborne Atmospheric
Measurements (FAAM) and Deutsches Zentrum fr Luft- und Raumfahrt e.V. (DLR) flight campaigns. The examples shown are for the Icelandic volcano Eyjafjallajökull (2010) due to the co-location of satellite data being greatest near the poles.

In this paper the fundamental instrument used in the analysis is described in section 2 followed by the introduction of the retrieval algorithm and forward model in section 3. The sensitivity of the retrieval to different error covariance matrices is discussed in section 4 and, after the results from comparisons of the retrieved IASI parameters with alternative data sources are
shown in section 5, conclusions are made in section 6.

## 2   IASI

IASI, on board the MetOp platforms, is a series of three identical Fourier transform spectrometers designed primarily to provide data to be assimilated for use in numerical weather prediction (NWP). The instrument is a Michelson interferometer covering the mid-infrared (IR) from $645-2760\,\mathrm{cm}^{-1}$ ($3.62-15.5\,\mu\mathrm{m}$) with a spectral resolution of $0.5\,\mathrm{cm}^{-1}$ (apodised) and
a pixel diameter at nadir of $12\,\mathrm{km}$. MetOp's sun-synchronous polar orbit and IASI's wide swath width means that global coverage is achieved twice daily with the day-time descending node overpass at 09:30 local time for IASI-A (Siméoni et al., 1997; Chalon et al., 2001; Hébert et al., 2004). Since aerosol fields have high spatial and temporal variability, regular views of the same area are essential to characterise plume evolutions. Therefore, IASI's characteristics make it a very useful tool for the



observation of larger aerosol particles (such as sand and volcanic ash). The work shown here uses IASI level 1c data obtained from the British Atmospheric Data Centre (BADC) archive.

## 3  Retrieval Method

### 3.1  Optimal Estimation Algorithm

A retrieval scheme using an optimal estimation framework has been developed to retrieve the properties of volcanic ash plumes. The scheme analyses the brightness temperature spectra from IASI in order to retrieve the following parameters: ash optical depth (at a reference wavelength of $550\,\mathrm{nm}$), ash effective radius ($\mu$m), ash plume top height (km) and surface temperature (K).

An ash detection method, based upon the trace gas detection method described by Walker et al. (2011) and applied to volcanic ash by Sears et al. (2013), flags IASI pixels for the presence of volcanic ash. In previous work, the presence of

volcanic $SO_2$ has been used as a proxy for the location of volcanic ash, therefore, pixels are also flagged for $SO_2$ in the same manner as Carboni et al. (2012) and the retrieval is subsequently calculated for pixels that are flagged to contain either a positive ash or $SO_2$ signal. The ash and $SO_2$ flags are produced in near-real time and the results are publicly available within 3 hours of measurement at http://www.nrt-atmos.cems.rl.ac.uk/.

For a detailed description of optimal estimation see Rodgers (2000), but essentially we define the measured spectra, $\mathbf{y}$, and

attempt to simulate it using the forward model, $\mathbf{F}(\mathbf{x}, \mathbf{b})$:

$$\mathbf{y} = \mathbf{F}(\mathbf{x}, \mathbf{b}) + \boldsymbol{\epsilon}, \tag{1}$$

where $\mathbf{x}$ is the state vector containing the parameters to be retrieved. All atmospheric properties needed by the forward model that are not retrieved are contained in $\mathbf{b}$ and $\boldsymbol{\epsilon}$ contains all the uncertainties associated with the retrieval.

The retrieval aims to find the most probable state of $\mathbf{x}$ by minimising the cost function, $\chi^2$, given by

$$\chi^2 = [\mathbf{y} - \mathbf{F}(\mathbf{x}, \mathbf{b})]^{\mathrm{T}} \mathbf{S}_\epsilon^{-1} [\mathbf{y} - \mathbf{F}(\mathbf{x}, \mathbf{b})] + [\mathbf{x} - \mathbf{x}_a]^{\mathrm{T}} \mathbf{S}_a^{-1} [\mathbf{x} - \mathbf{x}_a], \tag{2}$$

where $\mathbf{x}_a$ is the *a priori* value of the state vector and $\mathbf{S}_\epsilon$ and $\mathbf{S}_a$ are the measurement and *a priori* error covariance matrices respectively.

In order to find the the minimisation point of Eq. (2) the Levenberg-Marquardt-Press method is implemented, which numerically iterates the retrieval until a convergence criteria is satisfied, or a maximum number of iterations is reached. In the former

case the retrieval is considered to have converged; in the latter case, the retrieval is deemed to have failed and rejected. Full details of this implementation can be found in Rodgers (2000).

### 3.2  Assembling the Error Covariance Matrix

The measurement error covariance matrix, $\mathbf{S}_\epsilon$, is built up from an ensemble of difference spectra, capturing the variability between the IASI data and the radiative transfer model calculations. Each of the spectra is the residual between an IASI mea-

surement and a spectrum simulated using the forward model and co-located atmospheric profile data. Only scenes where there




is confidence that no volcanic signatures are present are included. This creates a generalised error covariance matrix containing not just the instrumental noise but also the spectral variability due to any inability for the forward model to correctly simulate the IASI measurements, for example: due to the presence of cloud, errors in the spectroscopy or errors in the atmospheric profiles. Assuming that the state of such variables are of no interest (in this problem) and the spectral signal of these variables are orthogonal to the ash signal, including these spectral signatures within the error covariance means there is no need for them to be retrieved nor their variance to be accounted for in the forward model of the atmosphere, thus allowing the problem to be simplified.

The elements of the error covariance are calculated using:

$$\mathbf{S}_{\epsilon}(i,j) = \langle [(\mathbf{y}_i - \mathbf{F}(\mathbf{x}_i)) - (\overline{\mathbf{y} - \mathbf{F}(\mathbf{x})})_i][(\mathbf{y}_j - \mathbf{F}(\mathbf{x}_j)) - (\overline{\mathbf{y} - \mathbf{F}(\mathbf{x})})_j] \rangle, \tag{3}$$

where $(\overline{\mathbf{y} - \mathbf{F}(\mathbf{x})})$ is the mean spectral difference between the measurement and a clear-sky simulation for each channel.

Given that the error covariance is relative to this mean residual, the input IASI spectrum to the retrieval must be adjusted to account for this bias. This produces a new cost function that must be minimised:

$$\chi^2 = [\mathbf{y} - \mathbf{F}(\mathbf{x}, \mathbf{b}) - \mathbf{c}]^{\mathrm{T}} \mathbf{S}_{\epsilon}^{-1} [\mathbf{y} - \mathbf{F}(\mathbf{x}, \mathbf{b}) - \mathbf{c}] + [\mathbf{x} - \mathbf{x}_a]^{\mathrm{T}} \mathbf{S}_a^{-1} [\mathbf{x} - \mathbf{x}_a], \tag{4}$$

where $\mathbf{c}$ is the mean residual.

Initially, when selecting the IASI scenes to include in the creation of the covariance matrix, all scenes with no volcanic ash signal were used. However, this produces a covariance with a large variance due to the large impact the presence of cloud has upon the spectral region chosen for the retrieval, i.e. any transparent window channels. In cloudy scenes, this makes the retrieval possible as the variation due to cloud is accounted for and, because of the large variance, the retrieval is able to converge. However, for a clear-sky scene, including such large variances allows the retrieval to appear to converge, albeit with a large uncertainty. Therefore, separate covariance matrices have been produced using solely either clear-sky scenes or cloudy scenes, where a cloudy scene is deemed to be where the window channel brightness temperature differs by more than $5\,\mathrm{K}$ from the ECMWF co-located surface temperature. These matrices shall henceforth be referred to as the 'clear' and 'cloudy' covariance matrices and can be seen in Fig. 1. Pictorial examples of the scenes each covariance applies to can be seen in Fig. 2. The clear-sky covariance also encompasses scenes for which there is a thin meteorological cloud beneath the plume that does not alter the window channel temperature significantly, whilst there is no covariance matrix that is able to cope with a thick meteorological cloud above the ash plume, meaning retrievals in these scenes are still challenging.

It must be noted that there are further error components that are not considered within the current covariance matrices that may be addressed in future work. These are the errors associated with the modelling of the plume, such as; assuming a plane parallel atmosphere, assuming that there is no leakage of radiation from the edges of the plume, assuming that the plume has only a single layer, and assuming the ash particles to be spherical and have a log-normal size distribution of fixed spread.



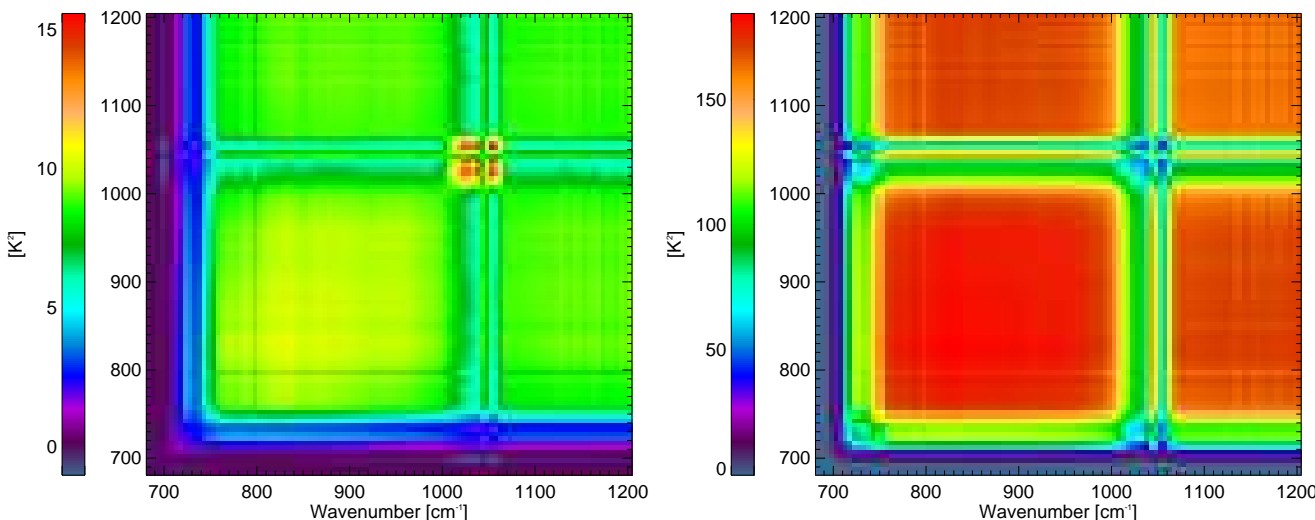

**Figure 1.** The measurement error covariance matrices created using IASI data from a) clear-sky scenes and b) cloudy scenes for the latitudinal band $30°$–$60°$ N

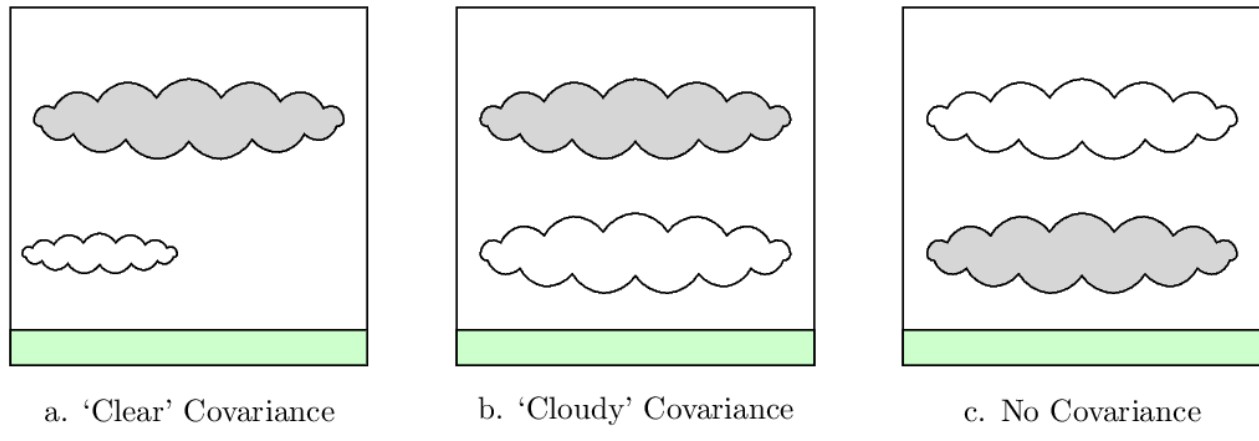

**Figure 2.** A pictorial form of the assumed scenes for each covariance matrix: a. A clear-sky scene, or a scene with only a thin layer of cloud beneath the ash plume, b. A cloudy scene, with a thick cloud below the ash plume, c. A cloudy scene for which we have no specific covariance, with a thick cloud above the ash plume.

## 3.3 Forward Model Description

Due to the computational intensity of the retrieval algorithm the forward model used to simulate the atmospheric conditions must be chosen to achieve a practical speed; for this a fast radiative transfer model is needed. The chosen model to simulate

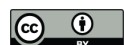



clear-sky radiances (i.e. containing gaseous absorbers but not cloud or aerosol/ash) is RTTOV, and its output is combined with an ash layer using the same scheme as that for the ORAC algorithm.

Standard atmospheric profiles are assumed within the forward model, except for temperature, pressure, altitude and water vapour. These profiles are obtained from ECMWF operational forecast data (European Centre for Medium-Range Weather Forecasts (2012

available for every six hours, and interpolated to the average time of the IASI orbit (or time of analysed section of orbit) to be processed and at the coordinates (latitude, longitude) of every IASI pixel. These profiles are then passed to the forward model. Future work may involve including the ECMWF ozone data as well, as the main ozone spectral absorption region (1000–1100 $\mathrm{cm}^{-1}$) can contain important information on ash type. Despite little trace gas information being included and RTTOV using default standard atmospheric conditions for most absorbers, the use of the generalised error covariance will account for

the variability of such species.

RTTOV provides the clear-sky top of atmosphere (TOA) radiance along with both upwelling and downwelling radiances at each altitude level. These can be used to formulate an overcast TOA radiance that includes a single layer of ash (or other broadband scatterer), which is assumed to be infinitely thin. Figure 3 shows schematically the interaction of the ash-atmosphere model. As per Thomas et al. (2009a), we define the following parameters to be:

– $R_\mathrm{o}^\uparrow$ The TOA clear-sky radiance

  – $R_\mathrm{al}^\uparrow$ The TOA radiance associated with the atmosphere above the ash layers

  – $R_\mathrm{bl}^\uparrow$ The upwelling radiance at the ash layer

  – $R_\mathrm{al}^\downarrow$ The downwelling radiance at the ash layer

  – $T_\mathrm{al}$ The transmission of the atmosphere above the ash layer

– $T_\mathrm{l}$ The transmittance of the ash layer

  – $B_\mathrm{s}$, $B_\mathrm{l}$ The Planck radiance of the surface and ash layer respectively

  – $\epsilon_\mathrm{s}$, $\epsilon_\mathrm{l}$ The emissivity of the surface and ash layer respectively

  – $R_\mathrm{l}$ The reflectivity of the ash layer

  – $P_\mathrm{l}$ The cloud pressure

$R_\mathrm{o}^\uparrow$, $R_\mathrm{al}^\uparrow$, $R_\mathrm{al}^\downarrow$, $R_\mathrm{bl}^{\uparrow atm}$ (the atmospheric contribution of $R_\mathrm{bl}^\uparrow$) and $T_\mathrm{al}$ are calculated efficiently within RTTOV. $R_\mathrm{bl}^\uparrow$ must account for the contribution of both the surface and the atmosphere below the ash layer to the upwelling radiance by $R_\mathrm{bl}^\uparrow = R_\mathrm{bl}^{\uparrow atm} + \epsilon_\mathrm{s} B_\mathrm{s} T_\mathrm{bl}$ where the total atmospheric transmission, $T_\mathrm{atm} = T_\mathrm{bl} T_\mathrm{al}$. The emissivity, reflectance and transmittance of the ash layer are functions of state vector elements, optical depth, $\tau$, effective radius, $r_\mathrm{eff}$, and plume top height, $h$ as well as the observation geometry. Computational efficiency is optimised by pre-computing these properties of the ash layer using DISORT

and storing the results in look-up-tables (LUTs), which are interpolated to the appropriate values.



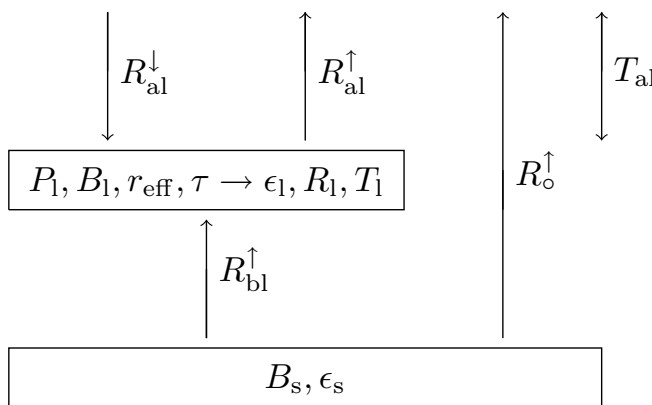

**Figure 3.** Schematic showing the atmospheric interactions simulated in the radiative transfer forward model

Given these quantities, and ignoring multiple reflections between the layer and the surface, the 'ash' TOA radiance, $R_\bullet^\uparrow$, can be expressed as

$$R_\bullet^\uparrow = R_{\mathrm{bl}}^\uparrow T_\mathrm{l} T_\mathrm{al} + B_\mathrm{l} \epsilon_\mathrm{l} T_\mathrm{al} + R_\mathrm{al}^\downarrow R_\mathrm{l} T_\mathrm{al} + R_\mathrm{al}^\uparrow, \tag{5}$$

where the terms on the right hand side correspond to, in order, the upwelling radiance below the ash layer transmitted by the layer and atmosphere above it, the emission from the ash layer, the reflected downwelling radiance above the ash layer and the upwelling radiance contribution from the atmosphere above the ash layer.

## 4  Error analysis/Sensitivity Study

An advantage of the optimal estimation framework is it provides a rigorous estimation of the uncertainty in the retrieved state. The *a posteriori* error covariance matrix, $\mathbf{S}_x$, can be written as

$$\mathbf{S}_x = (\mathbf{K}^\mathrm{T} \mathbf{S}_\epsilon^{-1} \mathbf{K} + \mathbf{S}_a^{-1})^{-1}, \tag{6}$$

where $\mathbf{K}$ is the Jacobian, which represents how the measurement spectrum is expected to change given a perturbation to the state. The diagonals of $\mathbf{S}_x$ provide the expected variance on the retrieved state vector elements and, hence, the square root of the diagonals give the uncertainty in each retrieved parameter. The optimal estimation retrieval produces the most probable values for; ash optical depth, particle effective radius, plume top height and surface temperature, each with associated uncertainties. Further, from these values and an assumed ash density, the ash mass in the plume can be derived.

An uncertainty analysis was performed using synthetic spectra (adding an ash plume to a reference clear atmosphere) to assess the sensitivity of the retrieved parameters to variations in the state. In the simulations the ash optical depth varied





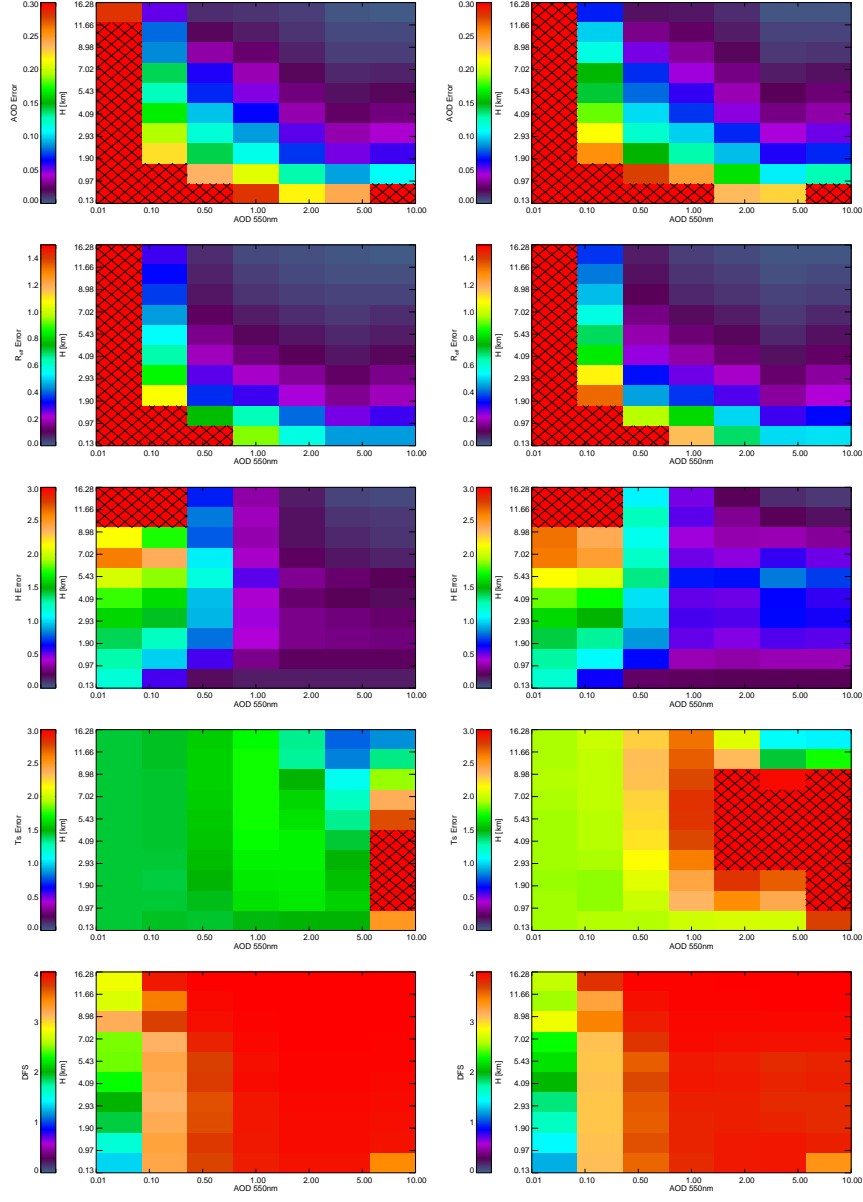

**Figure 4.** The uncertainties in the retrieved parameters and the DFS available in the retrieval using the clear (first column) and cloudy (second column) covariance matrices, shown as a funtion of ash optical depth (horizontally) and plume top altitude (vertically). From the top, the rows exhibit the uncertainties in ash optical depth, effective radius, plume top altitude, surface temperature and available DFS. Any value higher than the colour bar scale are shown hatched.

between 0.01 and 10 and the plume top altitude lay between $1000\,\mathrm{mb}$ and $100\,\mathrm{mb}$ ($\sim 0–16\,\mathrm{km}$). For the examples shown in Fig. 4 the effective radius and surface temperature are held constant at $2\,\mu\mathrm{m}$ and $291\,\mathrm{K}$ respectively. (Simulations were also





carried out varying these values but they are not shown here). The a priori uncertainty estimates used were $\pm 1$ for the logarithm of ash optical depth, $\pm 6\,\mu\text{m}$ for effective radius, $\pm 150\,\text{mb}$ for altitude and $\pm 20\,\text{K}$ for surface temperature. The results show the uncertainties in the retrieved parameters and the degrees of freedom (DFS) within the retrieval for different scenarios. All examples consider a 'local' error covariance matrix, $S_\epsilon$, which is computed using spectra located within the latitude band,

$30°$–$60°$ N, above the Icelandic plume region, which is calculated as described in section 3.2. The resultant uncertainties are shown for optimal estimation retrievals using both the clear and cloudy covariance matrices.

Using the clear covariance has consistently larger DFS available for all combinations of parameters, with optically thick plumes at lower altitudes having nearly 0.5 DFS more than the results using the cloudy covariance. The impact of this difference in DFS can be seen in the uncertainty in surface temperature, where the more optically thick plumes have a significantly larger

uncertainty. Interestingly, and perhaps unexpectedly, the surface temperature uncertainty improves at the highest altitudes. This is due to the substantially larger amount of atmosphere below a plume at $16\,\text{km}$ (as opposed to $8\,\text{km}$). This leads to an increase in the fraction of the total radiance across the window regions contributed by the atmosphere below the plume, and conversely, a decrease in the fraction of the total radiance that comes from the emission of the plume itself. Additionally, the region of low surface temperature uncertainty (high altitude and optically thick plume) coincides with the region with the lowest plume top

altitude uncertainty. Essentially, the more accurately we are able to retrieve the height of the plume, the more information that is available to improve the surface temperature estimate. The behaviour seen here was also exhibited in the retrieval of $SO_2$ by Carboni et al. (2012).

The retrieved altitude uncertainty is typically $< 1\,\text{km}$, with the uncertainty for optically thick plumes ($\text{AOD} > 1$) reducing to $< 0.5\,\text{km}$ when using the clear covariance. However, as the plume becomes optically thinner, and less information is available,

the retrieval tends towards the *a priori* value and the uncertainty estimate increases to become the *a priori* uncertainty.

For both ash optical depth and effective radius, the associated uncertainties have a similar pattern and are significantly lower for large values of optical depth. The smallest uncertainties occur for high altitude optically thick plumes, similar to both plume top height and surface temperature, as this is where the largest number of DFS are available. In contrast to the uncertainty in plume top altitude, the estimated uncertainties decrease as the height of the plume increases, with a maximum

expected uncertainty near the surface. This is most apparent at $\text{AOD} < 0.5$, with the maximum uncertainty occurring at the surface for the most optically thin plumes. At $\text{AOD} < 0.1$ it is observed that the uncertainty in effective radius and AOD reach $100\%$ and even higher in the latter case. Therefore, retrieval outputs of AOD values this small should be handled with caution.

The overarching behaviour of the expected uncertainty produced using both the clear and cloudy covariance matrices is the same, with the clear covariance producing consistently smaller uncertainty. It must be noted that this sensitivity study was

carried out assuming a clear atmosphere with no clouds (except for the volcanic ash plume) and therefore the clear covariance is expected to perform better due to the smaller variance it contains. If used for a cloudy scene, this smaller variance gives rise to the potential for a retrieval to have a high cost or fail to converge entirely, whereas the larger variance of the cloudy covariance matrix can account for the cloud beneath the plume and produce a better retrieval.

Ideally, the clear covariance matrix would be used for plume scenes where there is no meteorological cloud and the cloudy

covariance matrix would be used for scenes where meteorological cloud is detected. Cloud clearing of this manner is chal-



lenging and infeasible. Instead, a criteria is currently applied to the retrieval whereby it must pass a set of quality control tests. These ensure the output is sensible and realistic (e.g. the plume top altitude is not below the surface or the effective radius negative) but also only consider the retrieval a success if it converges within 10 iterations and the cost is below a specified threshold. Consequently, the retrieval is first carried out using the clear covariance matrix and then, if it fails to pass the quality control tests, the retrieval is further carried out using the cloudy covariance matrix. If the retrieval again fails to pass the quality control tests, it is discarded and deemed a failed retrieval.

## 5  Validation of retrieved parameters

An example of the retrieved plume properties and their associated uncertainties are shown in Fig. 5 for the morning of the $9^{th}$ May 2010 during the eruption of Eyjafjallajköull. This scene demonstrates the spatial consistency of the retrieval output and the histograms of the plume properties also show that the output values are distinctly moving away from the *a priori* values with the modal values for AOD, effective radius and height equaling $0.15$, $1\,\mu$m and $3.5\,$km respectively (the *a priori* values are $0.3$, $2\,\mu$m, $4.2\,$km respectively). The associated uncertainties are also promising, especially for the height product, which shows uncertainty of $\sim 1\,$km – the same as in the sensitivy study for synthetic spectra. This plume is optically thin with typical AOD values of $0.15$–$0.2$. In the sensitivity study, it was shown that large uncertainties were expected at low AOD. This is the case here for some pixels, however, the modal uncertainty of $\sim 0.1$ is less than the spread of retrieved values ($\sim 0.2$). The same behaviour is also observed in the effective radius and height products. In the following sections, the retrieval outputs are compared to ancillary data sources to ensure consistency with existing products. Further validation of this algorithm can be seen in Balis et al. (2016) and Corradini et al. (2016).

### 5.1  Aerosol Optical Depth: Comparison to MODIS

### 5.1.1  MODIS

MODIS instruments reside aboard the sun-synchronous orbiting NASA Terra (launched May 2002) and NASA Aqua (launched December 1999) satellites. Terra orbits at a 705 km altitude, with a period of 98.8 minutes, an inclination of $98.2°$ and a 10:30 equatorial crossing time on the descending node. Aqua orbits at a 705 km altitude, with a period of 98.4 minutes, an inclination of $98.1°$ and a 13:30 equatorial crossing time on the descending node. For this study we use Terra observations which are closer in time to IASI's 9:30 equatorial crossing time aboard Metop-A. With a cross track swath of 2330 km MODIS provides global coverage nearly every day. It has 36 bands from 0.41 to 15 $\mu$m at 250 m (0.645 and 0.858 $\mu$m), 500 m (5 bands from 0.469 and 2.130 $\mu$m) and 1 km (29 bands from 3.750 to 14.235 $\mu$m) spatial resolutions. For this study we use products derived from measurements aggregated to 1 km for all bands.





**Figure 5.** Example retrieval ouput from $9^{\text{th}}$ May 2010 during the eruption of Eyjafjallajköull. The left hand column presents the retrieved parameters; ash optical depth, effective radius and height. The centre column shows a histogram of the respective retrieved values with the *a priori* value indicated by the red dotted line and the right hand column contains the associated uncertainties for each parameter.

### 5.1.2 ORAC

The Optimal Retrieval of Aerosol and Cloud (ORAC) uses radiances measured from satellite based imaging radiometers, including MODIS, to retrieve aerosol, cloud, volcanic ash and surface properties. The retrieval algorithm is based on a long heritage of optimal estimation based on the work of Rodgers (2000) and authors cited within and has been applied by several

5 researchers for aerosol and cloud retrievals (Thomas et al., 2009b; Sayer et al., 2011; Poulsen et al., 2012) and more recently



for volcanic ash retrievals (McGarragh et al., 2016). For MODIS ash retrievals ORAC uses measurements of solar reflectance in bands 1, 2 and 6 (0.65, 0.86 and 1.64 $\mu$m) and thermal brightness temperature in bands 20, 27, 28, 29, 31, 32, 33, 34, 35 and 36 (3.8, 6.7, 7.3, 8.6, 11.0, 12.0, 13.3, 13.6, 13.9 and 14.2 $\mu$m). The primary retrieval parameters include ash optical thickness at 550 nm, effective radius of a log-normal ash particle size distribution, ash plume top pressure and surface temperature.

Parameters derived from these include the ash optical thickness at 11 $\mu$m, derived from the optical thickness at 550 nm and the effective radius; the ash plume top height and temperature, derived from the cloud top pressure and input meteorological profiles; and the ash mass loading, derived from the optical thickness at 550 nm, the effective radius and an assumed ash density of 2.6 (Neal et al., 1994).

### 5.1.3 Colocating the data

IASI and MODIS have very different fields of view and, hence, they must be co-located in order for a comparison to be carried out. The number of MODIS retrievals is far greater than that for IASI due to its better spatial resolution along the track. In order to compare the results, the MODIS data is aggregated onto the IASI resolution, i.e. all MODIS pixels within 6 km of the IASI pixel centre are used to formulate the average. Co-location is assumed if the IASI measurements and the MODIS measurements lie within 50 km and 1 hour of each other.

### 5.1.4 Results

A comparison of the AOD at 11 $\mu$m retrieved by both the IASI and MODIS algorithms is shown in Fig. 6. Although the retrievals provide their AOD output at 550 nm, these values are obtained by spectral extrapolation and the value of AOD at 11 $\mu$m is more appropriate for comparison as it lies within the actual wavenumber range for both instruments. The data shown is from the Eyjafjallajköull eruption in 2010 and only the retrievals that pass the imposed quality control measures for both

algorithms are shown. A further criteria was imposed upon the MODIS data that all of the data points averaged onto the IASI pixel resolution must be flagged as ash by the MODIS algorithm for the aggregated pixel to be used in the comparison. This is to ensure that we are comparing like with like.

Good correlation is observed between the two instruments with an RMS value of 0.46. This is especially true at lower values of AOD, where the RMS reduces to 0.2 (for AOD < 1) and 0.15 (for AOD < 0.5). As the AOD increases, the spread of the

data also increases with a tendency for MODIS to see a higher AOD than IASI. However, there is a time difference between the data points and therefore, the instruments may not be viewing the same part of the plume, despite attempts to minimise this. Hence, perfect agreement is not expected and the correlation seen is extremely encouraging.

### 5.2 Effective Radius: Comparison to aircraft measurements

### 5.2.1 Aircraft Description

Immediately following the Eyjafjallajköull eruption in 2010, it became clear that aircraft measurements (both in-situ and remote sounding) were needed in order to validate the ash dispersion forecasts. Two of the European aircraft deployed were the




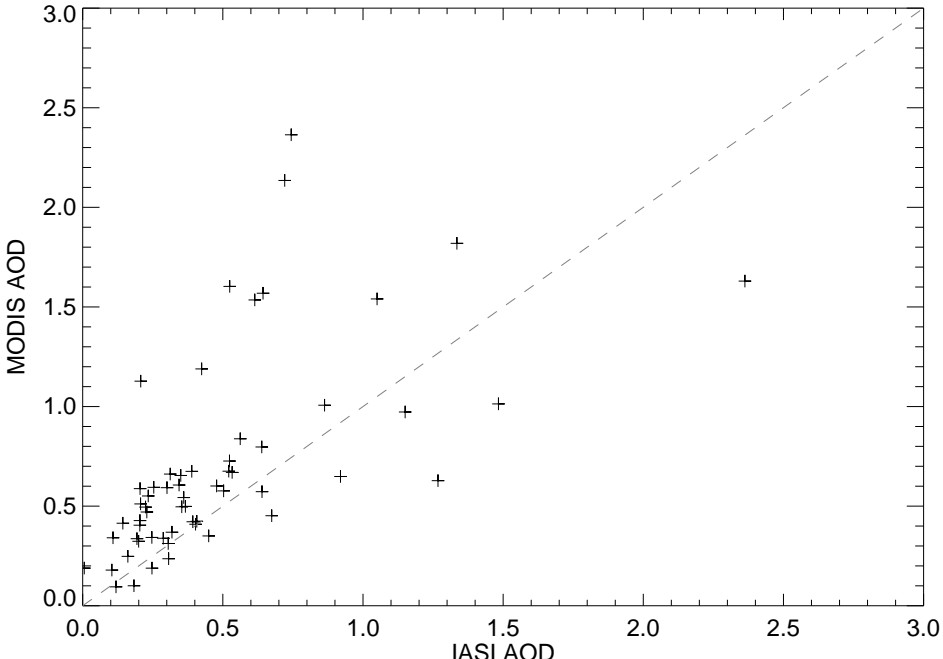

**Figure 6.** Comparison of AOD at $11\,\mu$m retrieved from IASI and MODIS during the Eyjafjallajköull eruption.

UK's BAe-146 FAAM aircraft (http://www.faam.ac.uk) and Germany's DLR Falcon aircraft (http://www.dlr.de). These aircraft are described in great detail elsewhere (see Marenco et al. (2011), Turnbull et al. (2012) and Newman et al. (2012) for FAAM aircraft; Schumann et al. (2011) for DLR aircraft) and therefore only a brief description is given here.

On board the FAAM aircraft were instruments capable of taking in situ and remotely sounded measurements. The in situ

5 observations come from two wing-mounted optical particle counters: a passive cavity aerosol spectrometer probe for particles with size distributions of diameter $0.1$–$3\,\mu$m and a cloud and aerosol spectrometer for particles of diameter $0.6$–$50\,\mu$m. Essentially these equate to fine and coarse mode aerosol respectively. The principal remotely sounded observations came from the on board lidar instrument; an ALS450 elastic backscatter lidar mounted to view in the nadir, which operates at a wavelength of $355\,$nm and has a footprint ranging from $7$–$11\,$km.

10 The DLR aircraft used the same instruments as the FAAM aircraft, however, the assumptions made in the calculation of the size distributions were different. Values for the optical properties (refractive index and shape) of the particles must be assumed as the response of the detectors is dependent upon these as well as the size (Turnbull et al., 2012). The DLR results assume spherical particles whereas the FAAM aircraft provide results for both spherical and irregular particles, with an additional result assuming the refractive indices of the DLR model, showing how the differing assumptions affect the results.

### 5.2.2 Results

Turnbull et al. (2012) provide in situ measurements of the Eyjafjallajköull volcanic ash cloud on $17^{\text{th}}$ May 2010 from both the FAAM and DLR flight campaigns. Despite no actual overlap in the flight paths of the aircraft, a worthwhile comparison is still possible and here we further compare to the IASI retrievals on the same day. Values for the geometric mean diameter

and standard deviation of the particle size distribution are given from both aircraft for both the fine and coarse particle modes. In order to compare these results to the retrieved IASI parameters, they must be converted into number weighted mean radius, $r_N$, by

$$r_N = \frac{D_g}{2} e^{-3\sigma^2},\qquad(7)$$

where $D_g$ is the geometric mean diameter by volume and $\sigma$ is the logarithm of the geometric standard deviation, $S$. Further,

due to IASI having sensitivity to both the fine and coarse modes, they are combined to calculate the effective radius, $r_{\text{eff}}$, using

$$r_{\text{eff}} = \frac{\sum_i m_i \exp\left[3\log r_{N,i} + \frac{9}{2}\sigma_i^2\right]}{\sum_i m_i \exp\left[2\log r_{N,i} + 2\sigma_i^2\right]},\qquad(8)$$

where the mixing ratios $m_i$ are the relative weight of each mode and a log-normal distribution is assumed. The size distributions obtained from the aircraft measurements can be seen in Table 1. Further, a histogram of the effective radius retrieved by IASI across all scenes containing the volcanic ash plume on the $17^{\text{th}}$ May can be seen in Fig. 7.

**Table 1.** The effective radius calculated for the size distributions measured on the $17^{\text{th}}$ May 2010

| Measurement Source | Effective Radius [$\mu$m] |
|---|---|
| FAAM Irregulars | 0.870 |
| FAAM Spheres | 0.977 |
| FAAM using DLR Refractive indices | 1.186 |
| DLR Spheres | 1.126 |

The values for the particle effective radius vary due to differing assumptions made in the calculations. The assumed type of size distribution in the retrieval and assumed shape in the aircraft calculations can impact the expected effective radius. The retrieved effective size distribution from IASI measurements is consistent with the values from the aircraft measurements, although slightly smaller. This is expected to be due, in part, to the sensitivities of the different instruments to different particle sizes, but also due to the IASI histogram including the full extent of the plume and, therefore, including regions where the

larger particles have been deposited out of the plume.




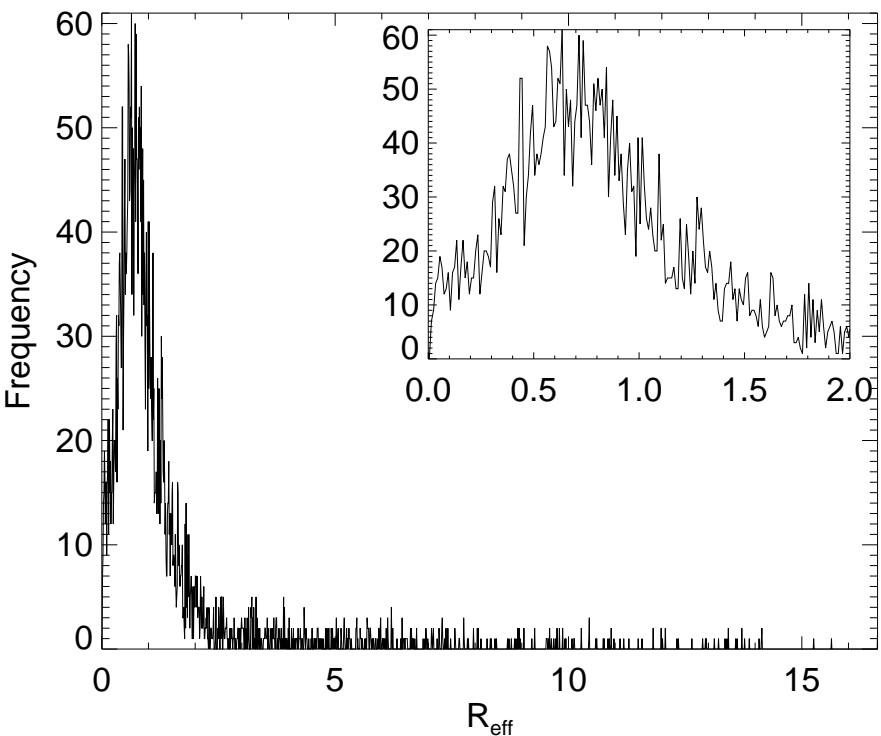

**Figure 7.** A histogram of the effective radius retrieved in the IASI scenes on 17[th] May 2010.

## 5.3 Plume altitude: Comparisons to CALIOP

### 5.3.1 CALIOP

CALIOP is the primary instrument onboard the Cloud-Aerosol Lidar and Infrared Pathfinder Satellite Observations (CALIPSO) satellite launched in May 2006. CALIPSO flies as part of the NASA Afternoon constellation (A-train) of satellites in a sun-

5 synchronous orbit with an equator crossing time (ascending) of 1:30 pm local solar time. With an orbit inclination of 98.2°, it provides a 16 day repeating cycle of coverage between 82° N and 82° S. CALIOP is a two-wavelength polarisation lidar viewing close to the nadir with a field of view of diameter 90 m at the ground. It measures the backscatter at two wavelengths, 532 nm and 1064 nm, with the returning signal to the 532 nm channel being polarised into the parallel and perpendicular components of the outgoing beam. The spatial resolution of CALIOP is nominally 30 m in the vertical and 335 m along track

10 (Winker David M. et al., 2009; Hunt et al., 2009).



### 5.3.2 Co-locating the data

Given the very differing footprints of IASI and CALIOP, they must be co-located in order to allow comparison. The frequency of IASI retrievals is far greater than that for CALIOP, however, CALIOP has far greater spatial resolution along the track. CALIOP observations of volcanic plumes have been identified using SEVIRI false colour images based on the infrared channels

at 8.7, 11 and 12 μm (Thomas and Siddans, 2015), where the backscatter profiles have been averaged vertically to a resolution of 250 m. In order to compare the results, the CALIOP data is smoothed onto the IASI resolution, i.e. all pixels become an average of the pixels within a 12 km spread. Co-location is assumed if the IASI measurements and the CALIOP measurements lie within 50 km and 1.5 hrs of each other. Where multiple CALIOP pixels satisfy this criteria for a selected IASI pixel, the CALIOP pixel closest in distance is chosen, under the assumption that the conditions will not vary much over the time period.

CALIOP produces atmospheric backscatter profiles for every pixel. However, the quantity required for validation is the cloud top height of the volcanic plume as this is the comparable ash property retrieved from IASI. Initially, the mean backscatter above 15 km is calculated for each CALIOP scene and is subtracted from the total backscatter. This removes any background backscatter leaving only the backscatter caused by the presence of clouds or the ash plume. For each CALIOP pixel the cumulative backscatter value is calculated descending through the atmosphere and the cloud top height is considered to be

at the altitude at which the atmospheric extinction passes a given threshold. For the purpose of this study, and given the manageable number of scenes considered, the threshold value is calculated individually for each scene and chosen to be the value that best matches the CALIOP image. An example of the derived cloud top height for a CALIOP scene is shown in Fig. 8.

### 5.3.3 Results

Due to the narrow swath of the CALIOP instrument, there are only a small number of coincidences between the two satellite datasets and there is no guarantee that the CALIOP track will intersect with the volcanic plume seen by IASI. This leads to only a few scenes available for comparison. Shown here are examples from the Eyjafjallajökull eruption in April/May 2010.

Figure 9 shows an overpass of Eyjafjallajökull on the $6^{th}$ May 2010. The colocation for this scene is good, with the CALIOP track directly crossing the retrieved IASI plume at latitudes above $55°$ N. It should be noted that co-location is much more

likely at high latitudes, near the poles. The scatter plot, shown in 10, comparing the retrieved IASI plume top height and the derived CALIOP plume top altitude, for this scene shows good agreement at low AOD but a significant underestimation for the optically thick pixels. This can be seen visually in the plot of plume top altitude as a function of latitude, overplotted on the CALIOP backscatter profiles, where the underestimated pixels can be clearly seen. It should be noted that these pixels occur in a region containing a vertical plume above the location of the volcano itself, which reaches up to the tropopause.

During this phase of the eruption the eruption column altitude was between 4 and 10 km (Stohl et al., 2011) as can be seen here in the CALIOP backscatter. Although the altitude retrieved by IASI does not match the CALIOP effective cloud top, the latitudinal location of the plume is correct, albeit the resultant altitude closer to the bottom of the plume. Also to note is that in this region the backscatter shows vertical breaks in the plume and hence, several layers of optically thick material.





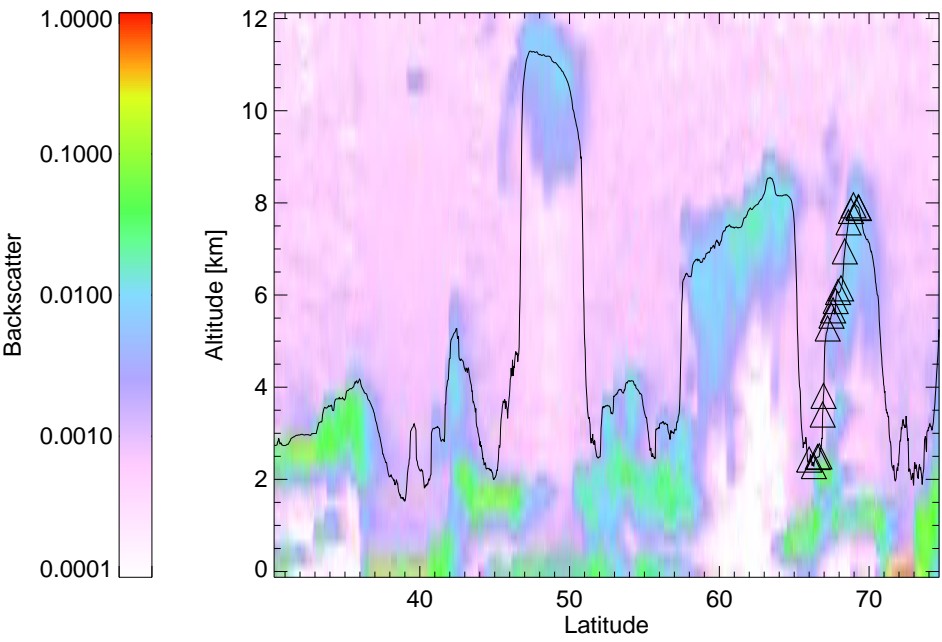

**Figure 8.** An example of the derived CALIOP cloud top heights are shown for an overpass of Grimsvötn on the 22$^{nd}$ May 2011. Pixels co-located with IASI are illustrated by triangles and the background shows the backscatter seen by CALIOP.

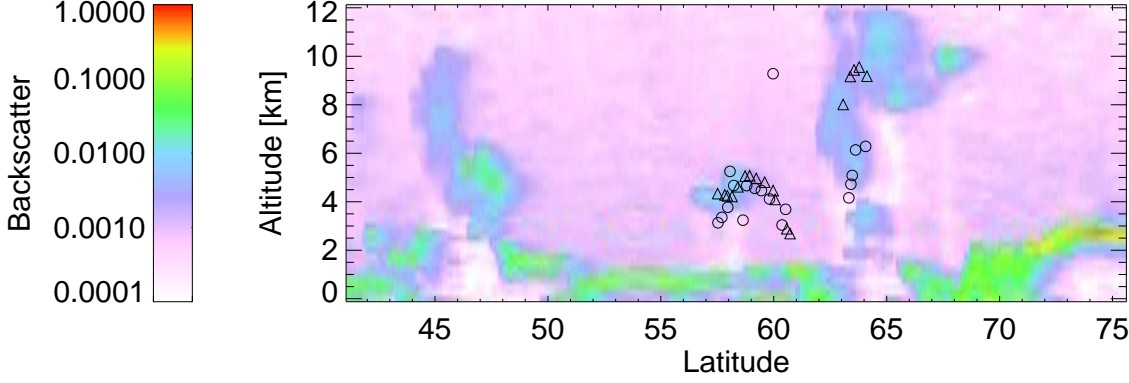

**Figure 9.** The results for an overpass of Eyjafjallajökull on the 6$^{th}$ May 2010. The derived CALIOP cloud top heights (triangles) and the retrieved IASI plume top altitude (circles) are overplotted onto the CALIOP backscatter.





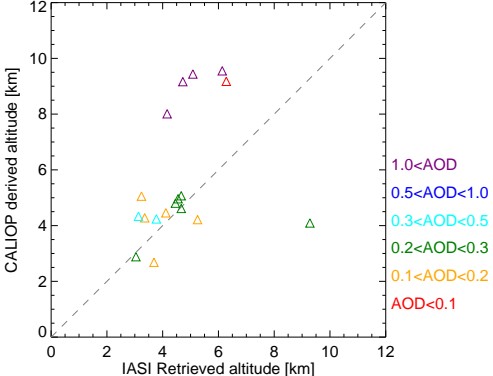
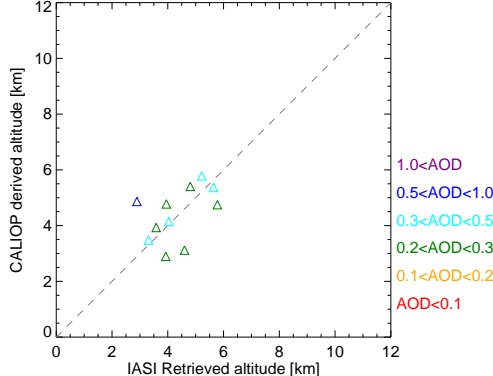

**Figure 10.** The results of comparisons between the retrieved IASI altitude and the derived CALIOP plume height for the Eyjafjallajökull eruption on the $6^{th}$ and $11^{th}$ May 2010

The retrieval algorithm assumes only one infinitely thin layer and therefore the retrieved altitude can be pulled closer to the surface to account for the lower layers. An alternative potential cause can be the *a priori* assumptions constraining the plume top altitude, however, relaxing the constraints did not improve the retrieval output. The outliers are reflected in the RMS value for the height comparison, which is 2.5 km. However, upon removing the optically thick outliers from the scene, this reduces the RMS difference to 0.9 km. The comparison for another well co-located scene is also shown in Fig. 10 for the $11^{th}$ May 2010, which also shows excellent agreement and an RMS value of less than 1 km.

Comparisons are not shown for all scenes individually, however, Fig. 11 shows the comparison for all points across all scenes. Some scenes have far fewer co-located pixels but do confirm that there is agreement between the CALIOP and IASI derived altitudes with the values largely occurring between 2 and 6 km. Visually, it can be seen that there are cases where the retrieval fails to fully capture the higher altitude plumes and there is an underestimation of the plume top height (as previously described), however, this is for only two of the scenes and given the time difference between the satellite overpasses, it is possible that the plume may have been transported vertically through the atmosphere during that time and therefore, small discrepancies are expected. Further, in general, these pixels tend to be optically thick, which may indicate that the IASI retrieval method is assuming a lower altitude and higher AOD in order to fit the measured spectra, whereas in reality the scene has a lower optical depth at higher altitude. This is symptomatic of the optimal estimation method and will be investigated further to reduce its occurrence. It should be noted that in many scenes sections of the plume are above layers of high backscatter, e.g. Fig. 9, which are low altitude meteorological cloud. In these cases the retrieval still performs well, although as stated, some underestimation in the plume top altitude is observed, caused by the multiple layers.



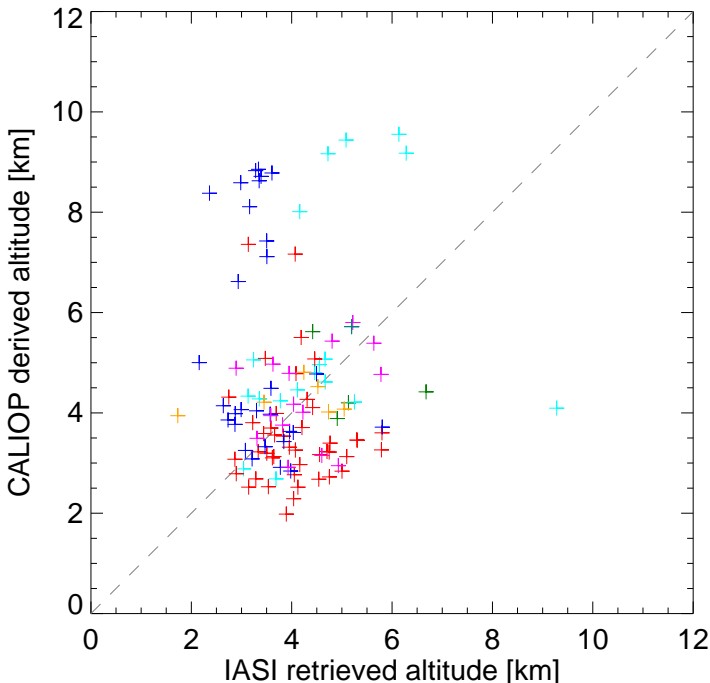

**Figure 11.** The results of comparisons between the retrieved IASI altitude and the derived CALIOP plume height for all scenes during the the Eyjafjallajökull eruption. The different colours indicate different scenes.

The results shown follow the criteria established in section 4; carry out the retrieval using the clear covariance matrix, then, if the retrieval fails (or does not pass the quality control) repeat using the cloudy covariance matrix. The robustness of this criteria was confirmed, by comparing the results shown in this paper to the results obtained using purely the cloudy covariance (not shown), with the latter performing worse.

## 6  Conclusions

A new optimal estimation scheme has been developed for the detection and characterisation of volcanic ash plumes using IASI measurements. Pixel-by-pixel estimates are derived for the properties of the volcanic ash; ash optical depth, effective radius, plume top altitude and surface temperature, with associated uncertainty estimates.

The measurement error covariance matrix is created using difference spectra, which are the residuals between IASI measurements and simulated spectra (calculated using RTTOV with ECMWF operational data). This ensures that all inaccuracies in the



simulation of the IASI spectrum, caused by lack of knowledge of the background atmospheric conditions (e.g. atmospheric profiles) or imperfections in the radiative transfer calculation (e.g. spectroscopy) are accounted for within the covariance matrix. Separate covariance matrices have been created using only clear-sky or cloudy scenes, where the latter contains the variance caused by the impact of meteorological cloud.

A sensitivity study has been carried out using both the clear-sky and cloudy covariance matrices, which showed that the clear covariance consistently produced smaller resultant uncertainty due to the smaller variance it contains. However, this only considers clear-sky synthetic spectra. Therefore, the criteria that is enforced first iterates the retrieval using the clear covariance matrix then, if the retrieval fails to pass the quality control tests (e.g. convergence), the retrieval is re-run using the cloudy covariance matrix. The uncertainty analysis demonstrates that the uncertainty in AOD, effective radius and plume top altitude

is higher for optically thin plumes, and for AOD and effective radius, this further increases as the plume nears the surface. In contrast to this, the uncertainty on plume top altitude decreases at lower altitudes.

    The results of comparisons between the retrieved volcanic ash properties and measurements from other instruments have been shown. The AOD has been show to have good agreement with retrievals carried out using the MODIS instrument onboard NASA TERRA. This is especially true at lower AOD (RMS $0.15$–$0.2$) with an increase in spread at increasing AOD (RMS

$0.46$). Aircraft campaigns during the Eyjafjallajökull eruption confirm that the retrieved size distribution from IASI is in line with the aircraft measurements, skewing towards slightly smaller particles due to viewing a larger area of the plume. Comparing the derived cloud top heights from CALIOP and retrieved IASI plume top heights further illustrate the robustness of the retrieval, with RMS values consistently less than $2\,\mathrm{km}$. Underestimation of the plume top altitude in optically thick pixels is observed, which is thought to be caused by the physical thickness of the plume, or the existence of multiple layers within the

plume, which are not accounted for in the forward model.

    Future work will aim to improve the current limitations within the retrieval and examine further ways to fully capture high plumes and account for the presence of cloud within the IASI scenes through the inclusion of another cloud (or ash) layer within the retrieval process (a two-layer forward model) and an improved selection of channels specifically chosen to minimise the impact caused by clouds.

**Data Availability**

The data shown in this paper for the Eyjafjallajökull eruption can be made available on request to the author. The IASI ash detection algorithm output is available in near real time at http://www.nrt-atmos.cems.rl.ac.uk.

*Acknowledgements.* RGG and EC were supported by the NERC Centre for Observation and Modelling of Earthquakes, Volcanoes, and Tectonics (COMET). The research leading to these results has received funding from the European Union's Seventh Framework Programme

(FP7/2007-2013) under grant agreement number 606738, APhoRISM project . The authors acknowledge funding from the NERC VANA-HEIM project NE/1015592/1 and the NERC SHIVA project NE/J023310/1. This study was funded as part of NERC's support of the National



Centre for Earth Observation. The authors would also like to acknowledge F. Marenco for providing additional FAAM aircraft data, that although not shown in this paper, has proved very useful in the study.





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
