# Peer review of "Retrieval of ash properties from IASI measurements"

_Atmospheric Measurement Techniques, 2016_

## Referee Comment (RC1) · Anonymous Referee #1 · 20 May 2016

General comments:

The authors present an interesting piece of work towards a better characterization of volcanic ash plumes from space. They use hyperspectral IASI measurements in the context of an optimal estimation scheme for assigning the most probable solution in terms of AOD, effective radius and ash layer altitude. Unfortunately the paper lacks a very important piece of information, which is essential to judge the quality and also the novelty of the presented approach. I was not able to find information about the methodology used by the authors to derive the optical properties of the ash, which are essential input to the radiative transfer calculations. The authors give hints that they assume spherical particles and mono-modal lognormal distributions. So I assume the optical properties have been calculated using Mie theory? With which code (there can be large differences!)? Which refractive indices have been used - and how have they

been spectrally interpolated (as there are not many refractive indices at the spectral resolution of IASI)? What parameters for the lognormal distributions have been assumed to end up with the presented effective radii? Without this information it is hard to understand what exactly are the capabilities of the method and which uncertainties may be hidden in the input assumptions. Consequently the manuscript requires major revisions before being considered for publication in AMT.

Specific comments:

p.1 l. 2: When the authors write "such as IASI", does that mean that the method as it is can be applied to AIRS or VIIRS? In that case not all IASI bands would be used (as AIRS and VIIRS lack significant parts of the specified range) and a description of the used IASI channels would be missing. Otherwise please make clear that the method has been developped for the use with IASI only.

p.1 l. 13: Please introduce abbreviations the first time they occur (RMS). Moreover, "RMS" is rather unspecific. What exactly is meant? Root Mean Square Difference (RMSD)? Root Mean Square Error (RMSE)?

p.1 l. 14: Please specify what exactly is meant by "low optical depth". AOD<0.1? AOD<0.5?

p.3 l.1: Typically the aerosol which can be detected by IASI is dust, not sand. "Sand" denotes particles >63$\mu$m, which are bound to the lowest part of the boundary layer due to the strong gravitational forces (unless in heavy sand storms, where turbulent forces can uplift even sand particles to larger altitudes - but they deposit rather fast after ceasing of the turbulent motions).

p.3 l.1: "IASI level 1c radiance data"

p. 4 l. 4: Is there any indication of the assumed orthoginality between ash signal and other signals in the spectra from theoretical considerations? If they are not orthogonal, the basic assumption behind the presented approach is at risk, so this orthogonality

should be somehow derived or at least be motivated in a convincing way (this does not mean that I do not believe in that orthogonality!).

p. 4 l. 7: I would like to see a small discussion about the assumed distribution of the brightness temperatures and especially about the assumed distribution of their uncertainties. I guess the authors assume the uncertainties being distributed normally, otherwise one could raise severe concerns about the validity of eq. (3). Such an assumption should be clearly stated.

p. 4 l. 15: "With no volcanic ash signal" or "with no volcanic ash detection"? That is significantly different!

p. 6 l. 8: Do the authors think there is sufficient information about ash type (what exactly do the authors mean by that word?) from below the O3 attenuation? I have some doubts...

p. 6. l. 8: What about SO2? Are there any (important) SO2 absorption bands within the selected wavenumber range?

p. 7 l. 15: There are a lot of other parameters to be assumed in order to derive the mass loading. First of all the assumed particle sphericity is a strong and definitely wrong assumption. That could be overcome by assuming an asphericity factor, which would impact on the volume estimation from the effective radius. With that regard - for nonspherical particles it must be defined if the effective radius is cross-section equivalent or volume equivalent, which can be totally different numbers. Then, in order to get to an estimate for the mass loading, the extinction efficiency needs to be known (estimated or assumed). For volcanic ash particles in the presented size range that is definitely not 2.0 ...

p. 9 l. 3: "Degrees of Freedom for Signal (DFS)" - There also exist a lot of other definitions and concepts of degrees of freedom.

p. 9 l. 6: I would suggest to shortly explain the concept of DFS for the readers not

familiar with it. Especially what we can learn from the numbers (by the way: why do the authors not show the DFS in Fig. 5?).

p. 9 l. 10: "Interestingly, and perhaps unexpectedly, the surface temperature uncertainty improves at the highest altitudes." To be honest, I do not understand this sentence. What exactly is at highest altitudes? The ash layer? I do not assume that surface temperature is at different altitudes? So please reformulate this sentence.

p. 9 l. 15: This is well known for quite a while now (for example S.A. Ackerman, 1997: Remote sensing aerosols using satellite infrared observations, J. Geophys. Res., 102, 17069-17079).

Section 5.1: Is the MODIS instrument described as input for ORAC? Then please make subsections 5.1.1-5.1.2 one subsection. Otherwise, if MODIS products are used, describe them (which algorithm, which collection, how they are aggregated).

p. 12 l. 5: How is 11$\mu$m AOD derived? Here again the description of the derivation of optical properties and basic assumptions is missing. Without that the reader is not able to understand how 11$\mu$m AOD and other ash layer parameters are derived.

p. 12 l. 20: It would be good to present the number of coincidences alongside.

p. 14 l. 4: Does that mean that for the aircraft data bimodal lognormal disrtibutions are assumed? It would be good to see the parameters for both modes along with the effective radii in table 1.

p. 15 l. 15: I have no hint from the manuscript where the authors derive this finding from. It would help to have the IASI derived effective radii averaged for the flight areas as well in table 1 (or at least mentioned in the text) or to have similar histograms as that in figure 7 from the aircraft data.

Figure 7: What is the bin size of the histogram? Is it really necessary to have such small bins (I assume the bin size is well below the assumed accuracy of the retrieval?)?

Figure 8: What exactly is colocated with what here? Are the black triangles IASI cloud top height? Does the CALIOP derived cloud top height include aerosol layers? More explanation is necessary.

p. 16 l. 20: How small is "small"? As before: it would be good to present numbers. Even if they are small: everone acknowledges that the conicidences are not widespread; providing these numbers does the manuscript no harm.

Figure 10 and 11: I would appreciate to have basic statistics (number of coincidences, correlation coefficient, bias, RMSD) together with the plots - either annotated to the plot or mentioned in the caption or the text.

p. 19 l. 10: I am not really convinced that this claim is true. What about the uncertainties of the ash optical properties? Where in the optimal estimation scheme are they reflected? Otherwise it is just not correct that all inaccuracies are accounted for.
* * *

---

## Referee Comment (RC2) · Anonymous Referee #2 · 6 Jun 2016

The manuscript describes a method for ash property retrievals using IASI measurements. The manuscript is well-written, but more details of the methodology and analysis are desirable and should be included before publication. Suggestions for improvements are given below.

- **Surface temperature retrieval**: In the abstract and elsewhere it is mentioned that the surface temperature is retrieved. However, in the manuscript no surface temperature retrievals are described. This should be one of the retrieved quantities that is easiest to compare with independent measurements or weather forecast models. Hence, please include a discussion and presentation of the surface temperature retrievals and comparison with relevant data.

- **Introduction, general comment:** A majority of earlier works on satellite ash

detection and retrieval use broad band instruments such as SEVIRI, MODIS, AVHRR etc. Please include a paragraph about what are the advantages and disadvantages with hyperspectral instruments. For example: hyperspectral instruments provide more spectral information and may thus potentially retrieve parameters that otherwise have to be assumed in retrievals using broad band instruments. On the other side, hyperspectral instruments typically have larger footprints than the broadband instruments. For example compare AVHRR and IASI which are on the same satellite. It should also be emphasized that you are retrieving the altitude of the plume height. The lack of plume height information is a major limitation in most split-window and similar techniques.

- **Page 2, lines 5-6**: Of the papers mentioned here, only the paper by Clarisse et al. (2010) use hyperspectral data, while the rest use broad band data. As this paper use IASI data it should be clearly stated that the other papers use the mentioned techniques on broad band data with limited spectral information. You may also want to mention that hyperspectral data may be used to retrieve the ash refractive index, see Ishimoto et al. (2016).

- **Page 2, line 20**: fr → für.

- **Page 3, line 9**: To make the manuscript self-contained, please include one or two sentences describing how the ash detection is done and IASI pixels flagged.

- **Page 16, line 9**: Please state which parameters are not retrieved but assumed and included in $b$. How does the assumed values of these parameters affect the retrieval error?

- **Page 3, line 23**: the the → the.

- **Page 3, line 24**: Please state your convergence criteria and maximum number of iterations.

- **Page 4, lines 4-5**: It is assumed that "these variables are orthogonal to the ash signal". May you please state what "these variables" are in order of importance? You mention clouds. Can you justify that ash clouds and for example liquid water clouds are orthogonal to each other using the difference in their optical properties?

- **Page 4, lines 20**: Please clarify if the forward model was cloudless also for the cloudy covariance matrix. Would it be possible to make covariance matrices for each effective cloud temperature and would you expect this to improve the retrieval?

- **Page 4, lines 24-26**: You mention clouds above and below the ash cloud. What about clouds at the same altitude as the ash cloud? And what about the presence of ice in the ash cloud itself? The latter is known to be a challenge, see for example Rose et al. (1995), Durant et al. (2008), Kylling (2016). Please discuss.

- **Page 6, line 13**: The ash cloud is assumed to be infinitely thin. Corradini et al. (2008) showed that ash cloud vertical extent have effect on the retrieved ash cloud optical properties. How realistic is the infinitely ash plume assumption and how does it affect your results? Is the error due to this assumption inluded in your error budget? If not, please make this clear in the manuscript.

- **Page 6, line 24**: $P_l$ is not used anywhere in the text. This line may be omitted.

- **Page 6, line 28**: Mention what ash size distribution is used and what parameters and values that describe it. Mention what ash type and refractive index that is used and include reference(s).

- **Page 8, line 1**: Please mention the wavenumber (wavelength) of the optical depths.

- **Page 9, line 4**: Please mention which longitudes are included in the "local" co-variance matrix.

- **Page 9, line 11-13**: I do not understand how this explains the decrease in temperature uncertainty. Please clarify. You may also possibly use the simplified model by Prata and Grant (2001) to explain the observed behaviour, see their Eqs. (2)-(5).

- **Page 9, line 24-26**: This behaviour may also possibly be explained by the simplified model of Prata and Grant (2001).

- **Page 10, line 4**: Please specify the threshold value.

- **Page 10, line 8**: Please mention what the average retrieved surface temperature including standard deviation. How does it compare to ECMWF values for the area?

- **Page 11-12, lines 2-8**: Do the MODIS and IASI retrievals use the same ash type and size distributions? If yes, please state so. If not, please state how any differences affect the comparison results.

- **Page 12, line 8**: What are the units of the number 2.6?

- **Page 12, line 12**: Several MODIS pixels cover one IASI pixel. Please mention how the MODIS ash optical propertis vary across the IASI pixels. This variability may be included as vertical error bars in Fig. 6.

- **Page 12, line 19**: Please mention what the "imposed quality controls" are.

- **Page 12, line 27**: Numbers for the "goodness" of the correlation may be obtained if fitting a straight line to the data.

- **Page 12, line 30**: Eyjafjallajköull → Eyjafjallajökull.

- **Page 14, lines 17-18**: You state that "The retrieved effective size distribution from IASI measurements is consistent with the values from the aircraft measurements, although slightly smaller." Here you state that you retrieve the effective size distribution from IASI measurements. Is this really so? Is it not the effective radius you retrieve based on an assumed size distribution? Please clarify.

- **Page 14, lines 15-20**: When comparing effective radii, please provide numbers for the IASI effective radius. This you may obtain by fitting a curve to the histogram in Fig. 7 and thus obtain an estimate of the IASI effective radius.

- **Page 16, lines 19**: Under **Results** also discuss Fig. 8. Also mention in the Introduction and Abstract that you include Grimsvotn data.

- **Page 16, lines 23**: What is implied by "The colocation for this scene is good"? Please quantify time and spatial differences.

- **Page 16, line 25**: 10 → Fig. 10.

- **Page 16, line 26**: Please quantify "good agreement".

- **Page 16, line 30**: Please beaware that the altitudes in Stohl et al. (2010) are derived from IASI and SEVIRI measurements using an inversion procedure. They only include the altitude of the fine ash that may be dispersed. Thus their use as a reference here is dubious. For the altitude of the plume above the volcanic vent the Arason et al. (2011) reference is maybe more appropriate.

- **Page 16, line 32**: It is not a big surprise that "the latitudinal location of the plume is correct". This statement may be omitted.

- **Page 17, Fig. 8**: What is shown by the solid line in the Figure?

- **Page 18, lines 14-15**: This could be due to the ash cloud being above an optically thick low altitude cloud, case b in Fig. 2. If the below cloud is optically thick

the retrieved surface temperature should represent that of the cloud and not the Earth's surface. Thus it would be interesting to know the retrieved surface temperatureds for these pixels and how they compare with the surface temperatures from for example ECMWF.

- **Page 20, lines 16**: You state "skewing towards slightly smaller particles due to viewing a larger area of the plume." However, I can not see that you have given evidence anywhere that the larger area is the reason. Yes, you speculate that this is the reason, but hard facts are needed to be able to firmly state this. Please clarify.

References

- P. Arason, G. N. Petersen and H. Bjornsson, Observations of the altitude of the volcanic plume during the eruption of Eyjafjallajökull, April-May 2010, Earth Syst. Sci. Data, 3, 9-17, doi=10.5194/essd-3-9-2011, 2011.

- Corradini, S., C. Spinette, E. Carboni, C. Tirelli, M. F. Buongiorno, S. Pugnaghi and G. Gangale, Mt. Etna tropospheric ash retrieval and sensitivity analysis using Moderate Resolution Imaging Spectroradiometer Measurements, J. of Applied Remote Sensing, 2, doi=10.1117/1.3046674, 2008.

- Durant, A. J., R. A. Shaw, W. I. Rose, Y. Mi and G. G. J. Ernst, Ice nucleation and overseeding of ice in volcanic clouds, Journal of Geophysical Research, 113, doi=10.1029/2007JD009064, 2008.

- Ishimoto, H., K. Masuda, K. Fukui, T. Shimbori, T. Inazawa, H. Tuchiyama, K. Ishii and T. Sakurai, Estimation of the refractive index of volcanic ash from satellite infrared sounder data, Remote Sensing of Environment, 174, 165 - 180, doi=http://dx.doi.org/10.1016/j.rse.2015.12.009, 2016.

[Figure]

- Kylling, A., Ash and ice clouds during the Mt Kelud February 2014 eruption as interpreted from IASI and AVHRR/3 observations, Atmospheric Measurement Techniques, 9, 2103–2117, doi=10.5194/amt-9-2103-2016, 2016.

- Prata, A. J. and I. F. Grant, Retrieval of microphysical and morphological properties of volcanic ash plumes from satellite data: Application to Mt Ruapehu, New Zealand, Q. J. R. Meteorol. Soc., 127, 2153-2179, 2001.

- Rose, W. I., Delene, D. J., Schneider, D. J., Bluth, G. J. S., Krueger, A. J., Sprod, I., McKee, C., Davies, H. L., and Ernst, G. G. J., Ice in the 1994 Rabaul eruption cloud: implications for volcano hazard and atmospheric effects, Nature, 375, 477-479, doi=10.1038/375477a0, 1995.

---

## Author Comment (AC1) · 1 Aug 2016

Response to Anonymous Referee #1

***Please note that all page and line references in the responses refer to the updated manuscript uploaded as supplementary material.***

General comments: The authors present an interesting piece of work towards a better characterization of volcanic ash plumes from space. They use hyperspectral IASI measurements in the context of an optimal estimation scheme for assigning the most probable solution in terms of AOD, effective radius and ash layer altitude. Unfortunately the paper lacks a very important piece of information, which is essential to judge the quality and also the novelty of the presented approach. I was not able to find information about the methodology used by the authors to derive the optical properties of the

ash, which are essential input to the radiative transfer calculations. The authors give hints that they assume spherical particles and mono-modal lognormal distributions. So I assume the optical properties have been calculated using Mie theory? With which code (there can be large differences!)? Which refractive indices have been used - and how have they been spectrally interpolated (as there are not many refractive indices at the spectral resolution of IASI)? What parameters for the lognormal distributions have been assumed to end up with the presented effective radii?

*The authors acknowledge that these are important factors that have been left out. The paper now includes from p.7 l. 25 'The emissivity, reflectance and transmittance of the ash layer are functions of the state vector elements, optical depth, $\tau$, effective radius, $r_{\text{eff}}$, and plume top height, $h$ as well as the observation geometry. Computational efficiency is optimised by pre-computing these properties of the ash layer using DISORT (Stamnes et al., 1988) and storing the results in look-up-tables (LUTs), which are linearly interpolated spectrally to the appropriate values. The spectral aerosol optical properties (extinction coefficient, single scattering albedo and the phase function) for ash are calculated using Mie theory (Grainger et al., 2004; code available at: http://www.eodg.atm.ox.ac.uk/MIE/index.html) and external mixing. The ash particles are assumed to be spherical with a mono-modal log normal aerosol size distribution, which has been shown to be a suitable representation of the size distribution of airborne volcanic ash (Wohletz et al., 1989). The distribution is characterized by a spread of $2$ (Wen and Rose, 1994b; Yu et al., 2002; Rybin et al., 2011; Pavolonis et al., 2013b) and the mode radius is translated to obtain different effective radii. The refractive index used in this paper is from measurements of ash from the Eyjafjallajökull eruption (Peters). These properties are calculated every $5\,cm^{-1}$ in the spectral range used by the retrieval, across a range of effective radii from $0.01$–$20\,\mu m$, to create the input for DISORT. Ignoring multiple reflections ...'*

Without this information it is hard to understand what exactly are the capabilities of the method and which uncertainties may be hidden in the input assumptions. Consequently the manuscript requires major revisions before being considered for publication in AMT.

Specific comments: p.1 l. 2: When the authors write "such as IASI", does that mean that the method as it is can be applied to AIRS or VIIRS? In that case not all IASI bands would be used (as AIRS and VIIRS lack significant parts of the specified range) and a description of the used IASI channels would be missing. Otherwise please make clear that the method has been developped for the use with IASI only.

***The method has currently only been applied to IASI. However, it could be adapted for use with other hyperspectral instruments, which would of course not necessarily use identical spectral bands. This has been made clearer in the text. The abstract now reads 'A new optimal estimation algorithm for the retrieval of volcanic ash properties has been developed for use with the Infrared Atmospheric Sounding Interferometer (IASI).' and at p.2 l.12 reads 'Presented here is a new optimal estimation algorithm for the retrieval of volcanic ash properties that has been developed for IASI, which could be further adapted for use with other hyperspectral satellite instruments.'***

p.1 l. 13: Please introduce abbreviations the first time they occur (RMS). Moreover, "RMS" is rather unspecific. What exactly is meant? Root Mean Square Difference (RMSD)? Root Mean Square Error (RMSE)?

***This has been corrected and defined as RMSE***

p.1 l. 14: Please specify what exactly is meant by "low optical depth". AOD$< 0.1$? AOD$< 0.5$?

***This has now been made more specific by inserting that it refers to AOD$< 0.1$.***

p.3 l.1: Typically the aerosol which can be detected by IASI is dust, not sand. "Sand"

denotes particles $> 63\mu$m, which are bound to the lowest part of the boundary layer due to the strong gravitational forces (unless in heavy sand storms, where turbulent forces can uplift even sand particles to larger altitudes - but they deposit rather fast after ceasing of the turbulent motions).

***This is an error in the text, it was intended to read 'dust' and has been corrected.***

p.3 l.1: "IASI level 1c radiance data"

***Corrected in the text.***

p. 4 l. 4: Is there any indication of the assumed orthoginality between ash signal and other signals in the spectra from theoretical considerations? If they are not orthogonal, the basic assumption behind the presented approach is at risk, so this orthogonality should be somehow derived or at least be motivated in a convincing way (this does not mean that I do not believe in that orthogonality!).

***The referee is correct, that orthogonality is required for the method to work. To demonstrate the difference between an ash signal and the non-ash signal we carried out a retrieval on a synthetic clear sky scene. If the signals were not orthogonal we would expect to retrieve ash values even though no ash was present. Such a retrieval gives an AOD$\sim 0.2$ a tiny (and negative) effective radius and the height of the plume is at the surface. The retrieval does not converge quickly and has a high cost due to attempting to fit an ash layer where there is none and therefore it would not pass the quality control. This indicates orthogonality but does not prove it. For the purposes of the paper the text has been edited to be more specific about the component retrieved. p.4 l.16 'Assuming that the state of such variables are of no interest (in this problem) and the spectral signal of these variables are orthogonal to the ash signal, including these spectral signatures within the error covariance means there is no need for them to be retrieved nor their variance to be accounted for in the forward model of the atmosphere, thus allowing the problem to be simplified. More specifically, the assumptions***

*in this method allow the retrieval of the orthogonal component of the retrieval parameters.'*

p. 4 l. 7: I would like to see a small discussion about the assumed distribution of the brightness temperatures and especially about the assumed distribution of their uncertainties. I guess the authors assume the uncertainties being distributed normally, otherwise one could raise severe concerns about the validity of eq. (3). Such an assumption should be clearly stated.

*Yes, it is assumed that the brightness temperature uncertainties are distributed normally. This has now been explicitly stated in the text.*

p. 4 l. 15: "With no volcanic ash signal" or "with no volcanic ash detection"? That is significantly different!

*This has been made clearer in the text. They are days that there is no known volcanic activity. p.4 l. 29 'Initially, when selecting the IASI scenes to include in the creation of the covariance matrix, only scenes from days with no known volcanic activity were used.'*

p. 6 l. 8: Do the authors think there is sufficient information about ash type (what exactly do the authors mean by that word?) from below the O3 attenuation? I have some doubts...

*The authors agree that this line is misleading and therefore it has been removed. It is hoped that ash type (i.e. ash derived from different magma composition) may be discerned from the extinction, which is at a maximum in the $O_3$ band.*

p. 6. l. 8: What about SO2? Are there any (important) SO2 absorption bands within the selected wavenumber range?

*There is an $SO_2$ band within the range used in the retrieval but the major absorbing channels are avoided in the channel selection, meaning there will be only a weak signal in those that are used.*

p. 7 l. 15: There are a lot of other parameters to be assumed in order to derive the mass loading. First of all the assumed particle sphericity is a strong and definitely wrong assumption. That could be overcome by assuming an asphericity factor, which would impact on the volume estimation from the effective radius. With that regard - for nonspherical particles it must be defined if the effective radius is cross-section equivalent or volume equivalent, which can be totally different numbers. Then, in order to get to an estimate for the mass loading, the extinction efficiency needs to be known (estimated or assumed). For volcanic ash particles in the presented size range that is definitely not 2.0 ...

*The authors agree that the ash mass calculation is not a simple one and realise that no reference was provided for our calculation. This has now been added (see p.9 l.3 '(see http:// eodg.atm.ox.ac.uk/ user/ grainger/ research/ aerosols.pdf )'). The authors also note, that the sphericity assumption is indeed not perfect but it has been shown that an assumption of spherical particles with a log-normal size distribution is a suitable representation of the size distribution of airborne volcanic ash (Wohletz et al., 1989). The reference has been added to text). Here we are merely stating that an ash mass can be calculated and this paper does not aim to prove the validity of the mass generated or any method to do so. Therefore, it is felt that further description is not needed.*

p. 9 l. 3: "Degrees of Freedom for Signal (DFS)" - There also exist a lot of other definitions and concepts of degrees of freedom.

*This has been corrected in the text.*

p. 9 l. 6: I would suggest to shortly explain the concept of DFS for the readers not familiar with it. Especially what we can learn from the numbers (by the way: why do the authors not show the DFS in Fig. 5?).

*A description has been added in the text: see p.9 l.10 'The results show the uncertainties in the retrieved parameters and the Degrees of Freedom for Signal*

[Figure]

*(DFS) within the retrieval for different scenarios. The DFS is a figure of merit that expresses the information contained in a retrieval by compressing the information within the retrieval error covariance matrix into a single scalar quantity. Essentially, it provides the number of independent pieces of information available in an estimate of the state.' The authors have not shown the DFS in Fig. 5 as it was not deemed necessary given the information provided regarding the DFS in section 4.*

p. 9 l. 10: "Interestingly, and perhaps unexpectedly, the surface temperature uncertainty improves at the highest altitudes." To be honest, I do not understand this sentence. What exactly is at highest altitudes? The ash layer? I do not assume that surface temperature is at different altitudes? So please reformulate this sentence.

*The sentence has been restructured to make sense. The altitude referred to the height of the assumed ash layer. p.9 l.20 'the surface temperature uncertainty improves when the ash layer is at the highest altitudes.'*

p. 9 l. 15: This is well known for quite a while now (for example S.A. Ackerman, 1997: Remote sensing aerosols using satellite infrared observations, J. Geophys. Res., 102, 17069-17079).

*The reference has been inserted.*

Section 5.1: Is the MODIS instrument described as input for ORAC? Then please make subsections 5.1.1-5.1.2 one subsection. Otherwise, if MODIS products are used, describe them (which algorithm, which collection, how they are aggregated).

*The MODIS data is used as an input to the ORAC algorithm so the sections have been combined and re-titled 'MODIS Retrieval Method'*

p. 12 l. 5: How is 11$\mu$m AOD derived? Here again the description of the derivation of optical properties and basic assumptions is missing. Without that the reader is not able to understand how 11$\mu$m AOD and other ash layer parameters are derived.

*A more detail description of the ORAC algorithm has been included including information on the optical properties and how the 11μm AOD is derived. p.13 l.10 'For MODIS ash retrievals ORAC uses measurements of solar reflectance in bands 1, 2 and 6 (0.65, 0.86 and 1.64 μm) and thermal brightness temperature in bands 20, 27, 28, 29, 31, 32, 33, 34, 35 and 36 (3.8, 6.7, 7.3, 8.6, 11.0, 12.0, 13.3, 13.6, 13.9 and 14.2 μm). The primary retrieval parameters include ash optical thickness at $550$ nm, effective radius of a log-normal ash particle size distribution, ash plume top pressure and effective radiating temperature. Parameters derived from these include the ash optical thickness at 11 μm, derived from the optical thickness at $550$ nm and the effective radius; the ash plume top height and temperature, derived from the cloud top pressure and input meteorological profiles; and the ash mass loading, derived from the optical thickness at $550$ nm, the effective radius and an assumed ash density of 2.6 g cm$^{-3}$ (Neal et al., 1994).*

*The ash particles are assumed to be spherical with a log-normal size distribution and the size distribution averaged spectral optical properties (extinction coefficient, single scattering albedo and phase function are calculated using Mie theory. Since the width of the distribution is not a retrieval parameter it must be assumed and a standard deviation of 2.0 is the value adopted for the ORAC retrieval. The complex index of refraction must also be assumed for which we use values measured from Eyjafjallajökull ash samples (Peters). These properties are the same as those assumed in the IASI retrieval.*

*The ash optical properties are further used as input to the plane parallel radiative transfer solver DISORT to compute scalar spectral reflection, transmission and emission operators used in a "fast" forward model, details of which are described in McGarragh et al. (2016), and stored in LUTs as a function of the retrieved 0.55-μm optical thickness and effective radius, in addition to the solar and satellite geometry. The optical thickness at 11 μm is obtained directly from the ratio of the extinction coefficient at 11 μm to that at 0.55 μm. The ash plume*

*is assumed to be infinitely thin geometrically allowing for a full decoupling of the ash radiative transfer from that of the clear-sky for which the transmittance and emission are computed with RTTOV from meteorological pressure, temperature, humidity and ozone profiles from the ECMWF ERA-Interim reanalysis product (Dee et al., 2011). Molecular (Rayleigh) scattering is computed according to Bates (1984) from the pressure and temperature profiles. Finally, the surface is characterized with a bidirectional reflectance distribution function (BRDF) for both land (Schaaf et al., 2002) and ocean (Sayer et al., 2010). Specific details regarding the sources of uncertainty are discussed in Thomas et al. (2009b), Sayer et al. (2011) and McGarragh et al. (2016).'*

p. 12 l. 20: It would be good to present the number of coincidences alongside.

*This number has been added to the text. p.14 l.10 'The data shown is from the Eyjafjallajköull eruption in 2010 and only the retrievals that pass the imposed quality control measures for both algorithms are shown (73 coincidences).'*

p. 14 l. 4: Does that mean that for the aircraft data bimodal lognormal disrtibutions are assumed? It would be good to see the parameters for both modes along with the effective radii in table 1.

*Yes the aircraft observed bimodal distributions and fit the geometric mean diameter, standard deviation and relative weights. These values are taken from Turnbull et al. (2012) and have been added to the table.*

p. 15 l. 15: I have no hint from the manuscript where the authors derive this finding from. It would help to have the IASI derived effective radii averaged for the flight areas as well in table 1 (or at least mentioned in the text) or to have similar histograms as that in figure 7 from the aircraft data.

*The authors are not in possesion of the data to provide histograms of the aircraft measurements. However, the mode value of effective radius for the entire plume*

*observed by IASI has been added to the table for comparison. It was tested lim-
iting the data to only within 50km of the aircraft track but this made no difference
to the observed mode.*

Figure 7: What is the bin size of the histogram? Is it really necessary to have such small
bins (I assume the bin size is well below the assumed accuracy of the retrieval?)?

*The authors agree that the bin size was too small and have increased it to $0.1\,\mu m$*

Figure 8: What exactly is colocated with what here? Are the black triangles IASI cloud
top height? Does the CALIOP derived cloud top height include aerosol layers? More
explanation is necessary.

*It has been made clearer in the caption that the triangles are the CALIOP derived
height at the locations that are coincident with an IASI measurement. Fig. 8
Caption: 'An example of the derived CALIOP cloud top heights are shown (as
the solid line) for an overpass of Grimsvötn on the $22^{nd}$ May 2011. The CALIOP
derived height at locations co-located with IASI pixels are illustrated by triangles
and the background shows the backscatter seen by CALIOP.'*

p. 16 l. 20: How small is "small"? As before: it would be good to present numbers. Even
if they are small: everone acknowledges that the conicidences are not widespread;
providing these numbers does the manuscript no harm.

*The numbers have been added and the sentence now reads: 'Due to the narrow
swath of the CALIOP instrument, there are only 8 coincidences (119 pixels in
total) between the two satellite datasets where the CALIOP track intersects with
the volcanic plume seen by IASI.'*

Figure 10 and 11: I would appreciate to have basic statistics (number of coincidences,
correlation coefficient, bias, RMSD) together with the plots - either annotated to the
plot or mentioned in the caption or the text.

*Further statistics have been added to the text. p.19 l.3 '...The outliers are re-*

*flected in the RMSE value for the height comparison, which is 2.5 km ($r = 0.31$).
However, upon removing the optically thick outliers from the scene, this reduces
the RMSE difference to 0.8 km and increases the correlation to $r = 0.41$. The
comparison for another well co-located scene is also shown in Fig. 10 for the
$11^{th}$ May 2010, which again shows good agreement with $r = 0.46$ and an RMSE
value of $0.9$ km.*

*Comparisons are not shown for all scenes individually, however, Fig. 11 shows
the comparison for all points across all scenes. Some scenes have far fewer
co-located pixels but do confirm that there is agreement between the CALIOP
and IASI derived altitude range with the values largely occurring between $2$ and
$6$ km. Despite good correlation in individual scenes, it is very low for all pixels,
$r = 0.12$, with RMSE of $2.1$ km. Visually, it can be seen that there are cases
where the retrieval fails to fully capture the higher altitude plumes and there is an
underestimation of the plume top height (as previously described), however, this
is for only two of the scenes and given the time difference between the satellite
overpasses,...'*

p. 19 l. 10: I am not really convinced that this claim is true. What about the uncertainties of the ash optical properties? Where in the optimal estimation scheme are they
reflected? Otherwise it is just not correct that all inaccuracies are accounted for.

*The text is perhaps misleading and has been altered to correct for this. The
uncertainties accounted for are those in the radiative transfer forward model,
not the ash optical properties used in the Mie code. p.22 l.2 '... This ensures
that all inaccuracies in the radiative transfer modelling of the IASI spectrum,
caused by lack of knowledge of the background atmospheric conditions
(e.g. atmospheric profiles) or imperfections in the radiative transfer calculation
(e.g. spectroscopy) are accounted for within the covariance matrix. Separate
covariance matrices have been created using only clear-sky or cloudy scenes,
where the latter contains the variance caused by the impact of meteorological*

***cloud. It should be noted that this does not account for errors in the ash optical properties. ...'***

Please also note the supplement to this comment:
http://www.atmos-meas-tech-discuss.net/amt-2016-143/amt-2016-143-AC1-supplement.pdf

————————————————————

---

## Author Comment (AC2) · 1 Aug 2016

Response to Anonymous Referee #2

*Please note that all page and line references in the responses refer to the updated manuscript uploaded as supplementary material.*

The manuscript describes a method for ash property retrievals using IASI measurements. The manuscript is well-written, but more details of the methodology and analysis are desirable and should be included before publication. Suggestions for improvements are given below.

- Surface temperature retrieval: In the abstract and elsewhere it is mentioned that the surface temperature is retrieved. However, in the manuscript no surface tem-

[Figure]

perature retrievals are described. This should be one of the retrieved quantities that is easiest to compare with independent measurements or weather forecast models. Hence, please include a discussion and presentation of the surface temperature retrievals and comparison with relevant data.

*The authors have not shown the output of the 'surface temperature' retrieval as there is no resource that it can be easily compared to. In reality, the retrieved parameter is the 'effective radiating temperature' not surface temperature and therefore a comparison against, for example, ECMWF data would be meaningless. It is used in the retrieval to help ground the retrieval but the output is not directly applicable to a real quantity. The authors feel that, in retrospect, it is perhaps misleading to refer to it as a surface temperature and therefore have replaced each occurrence with 'effective radiating temperature'.*

- Introduction, general comment: A majority of earlier works on satellite ash detection and retrieval use broad band instruments such as SEVIRI, MODIS, AVHRR etc. Please include a paragraph about what are the advantages and disadvantages with hyperspectral instruments. For example: hyperspectral instruments provide more spectral information and may thus potentially retrieve parameters that otherwise have to be assumed in retrievals using broad band instruments. On the other side, hyperspectral instruments typically have larger footprints than the broadband instruments. For example compare AVHRR and IASI which are on the same satellite. It should also be emphasized that you are retrieving the altitude of the plume height. The lack of plume height information is a major limitation in most split-window and similar techniques.

*The following comparison has been added to the text. p.2 l.6 'These methods have been applied to both hyperspectral and broad band satellite instruments, each of which have advantages and disadvantages. For example, IASI, on board MetOp-A, has a wealth of spectral information with over*

*8000 wavenumber channels allowing the potential to retrieve many parameters. Whereas the Advanced Very-High Resolution Radiometer, AVHRR, a broad band instrument, has only 6 channels to extract information from meaning more assumptions must be made about the state. However, the spatial coverage of AVHRR is much greater than IASI with a footprint of $\sim 1$ km compared to IASI's 12km footprint giving far more measurements within a volcanic plume. Presented here is a new optimal estimation algorithm for the retrieval of volcanic ash properties that has been developed for IASI to take advantage of its spectral information, which could be further adapted for use with other hyperspectral satellite instruments.'*

- Page 2, lines 5-6: Of the papers mentioned here, only the paper by Clarisse et al. (2010) use hyperspectral data, while the rest use broad band data. As this paper use IASI data it should be clearly stated that the other papers use the mentioned techniques on broad band data with limited spectral information. You may also want to mention that hyperspectral data may be used to retrieve the ash refractive index, see Ishimoto et al. (2016).

  *The authors believe this point has now been addressed in the above paragraph.*

- Page 2, line 20: fr → für.

  *This has been corrected in the text.*

- Page 3, line 9: To make the manuscript self-contained, please include one or two sentences describing how the ash detection is done and IASI pixels flagged.

  *The following description has been added to the text. p.3 l.17 '... flags IASI pixels for the presence of volcanic ash. The detection procedure looks for departures in a spectrum from an expected background covariance. An ensemble training set of IASI data, assumed to contain no extraordinary*

*ash concentrations, is used to create a generalised error covariance matrix that contains the spectral variability caused by interfering trace species and clouds as well as the IASI instrument noise. A least squares fit retrieval is carried out to retrieve the ash optical depth at three assumed altitudes; $400$ mb, $600$ mb and $800$ mb. The pixel is flagged if the ash optical depth at any of the altitudes passes a given threshold. In previous work, the presence of volcanic $SO_2$ has been used as a proxy ...'*

- Page 16, line 9: Please state which parameters are not retrieved but assumed and included in b. How does the assumed values of these parameters affect the retrieval error?

   *The parameters that are not retrieved, such as temperature profile, gas profiles, spectroscopy etc. are all mentioned in the following section regarding the error covariance matrix, which explains why they are not retrieved and how their uncertainties are accounted for within the covariance. The authors therefore feel that a repetition of that here is unnecessary.*

- Page 3, line 23: the the → the.

   *This has been corrected in the text.*

- Page 3, line 24: Please state your convergence criteria and maximum number of iterations.

   *These numbers have been added to the text. p4. l.5 '...the Levenberg-Marquardt-Press method is implemented, which numerically iterates the retrieval until a convergence criteria is satisfied (a positive or negative change in the cost of 1), or a maximum number of iterations is reached (default is 10)...'*

- Page 4, lines 4-5: It is assumed that "these variables are orthogonal to the ash signal". May you please state what "these variables" are in order of importance?

You mention clouds. Can you justify that ash clouds and for example liquid water clouds are orthogonal to each other using the difference in their optical properties?

*The method itself requires orthogonality for it to work. To demonstrate the difference between an ash signal and the non-ash signal we carried out a retrieval on a synthetic clear sky scene. If the signals were not orthogonal we would expect to retrieve ash values even though no ash was present. Such a retrieval gives an AOD$\sim 0.2$ a tiny (and negative) effective radius and the height of the plume is at the surface. The retrieval does not converge quickly and has a high cost due to attempting to fit an ash layer where there is none and therefore it would not pass the quality control. This indicates orthogonality but does not prove it. For the purposes of the paper the text has been edited to be more specific about the component retrieved. p.4 l.16 'Assuming that the state of such variables are of no interest (in this problem) and the spectral signal of these variables are orthogonal to the ash signal, including these spectral signatures within the error covariance means there is no need for them to be retrieved nor their variance to be accounted for in the forward model of the atmosphere, thus allowing the problem to be simplified. More specifically, the assumptions in this method allow the retrieval of the orthogonal component of the retrieval parameters.'*

- Page 4, lines 20: Please clarify if the forward model was cloudless also for the cloudy covariance matrix. Would it be possible to make covariance matrices for each effective cloud temperature and would you expect this to improve the retrieval?

*This has been clarified in the text. The forward model is always assumed to be cloudless during the retrieval (only an ash layer) with the uncertainty due to a meteorological cloud contained in the covariance matrix - which is the difference between clear sky forward model simulations and potentially*

*cloudy IASI scenes. p.5 l.6 'In both instances the forward model assumes a clear-sky scene scene.'*

*It would be possible to create different covariances for different cloud temperatures. However, within the retrieval it would be challenging to know which one to use, unless we have some a priori information on the cloud altitude (as both ash and meteorological clouds will decrease the brightness temperature). There is potential to implement them using data from either a previous pixel or an alternative source but this would entail a lot of work to make a large change to the retrieval and is beyond the scope of this paper.*

- Page 4, lines 24-26: You mention clouds above and below the ash cloud. What about clouds at the same altitude as the ash cloud? And what about the presence of ice in the ash cloud itself? The latter is known to be a challenge, see for example Rose et al. (1995), Durant et al. (2008), Kylling (2016). Please discuss.

*The authors agree with the referees remark regarding ice. It provides a large challenge and some discussion has now been added to the text. p.5 l.7 'The clear-sky covariance also encompasses scenes for which there is a thin meteorological cloud beneath the plume that does not alter the window channel temperature significantly, whilst there is no covariance matrix that is able to cope with a thick meteorological cloud above the ash plume, meaning retrievals in these scenes are still challenging. The covariance used in scenes where meteorological cloud is at the same altitude as the ash plume will depend upon the optical thickness of the cloud and the retrieved ash optical depth is expected to be an underestimate of the actual ash plume properties. Further challenges caused by the presence of ice in the ash plume due to the similarity in their spectral signatures are well known (Rose W. I. et al., 1995; Durant et al., 2008; Kylling, 2016). Some of their variability will have been captured in the covariance matrices. How-*

*ever, if the ash particles have become coated in ice, the optical properties are changed and the retrieval may underestimate the quantity of ash. Further work will look to better distinquishing the ash and ice cloud signitures. '*

- Page 6, line 13: The ash cloud is assumed to be infinitely thin. Corradini et al. (2008) showed that ash cloud vertical extent have effect on the retrieved ash cloud optical properties. How realistic is the infinitely ash plume assumption and how does it affect your results? Is the error due to this assumption inluded in your error budget? If not, please make this clear in the manuscript.

  *The authors are aware that the infinitely thin ash plume is a large assumption and will induce errors in the retrieval. However, for our retrieval method it must be assumed to be infinitely thin geometrically to allow for a full decoupling of the ash radiative transfer from that of the clear-sky radiative transfer. It is already stated in the manuscript that the error due to this is currently not taken into account in our error covariance matrix p.5 l.17 'It must be noted that there are further error components that are not considered within the current covariance matrices that may be addressed in future work. These are the errors associated with the modelling of the plume, such as; assuming a plane parallel atmosphere, assuming that there is no leakage of radiation from the edges of the plume, assuming that the plume has only a single layer, and assuming the ash particles to be spherical and have a log-normal size distribution of fixed spread. '*

- Page 6, line 24: PI is not used anywhere in the text. This line may be omitted.

  *The text has been removed*

- Page 6, line 28: Mention what ash size distribution is used and what parameters and values that describe it. Mention what ash type and refractive index that is used and include reference(s).

*In response to this and other comments the paper now includes a more detailed description from p.7 l. 25 'The emissivity, reflectance and transmittance of the ash layer are functions of the state vector elements, optical depth, $\tau$, effective radius, $r_{\mathrm{eff}}$, and plume top height, $h$ as well as the observation geometry. Computational efficiency is optimised by pre-computing these properties of the ash layer using DISORT (Stamnes et al., 1988) and storing the results in look-up-tables (LUTs), which are linearly interpolated spectrally to the appropriate values. The spectral aerosol optical properties (extinction coefficient, single scattering albedo and the phase function) for ash are calculated using Mie theory (Grainger et al., 2004; code available at: http://www.eodg.atm.ox.ac.uk/MIE/index.html) and external mixing. The ash particles are assumed to be spherical with a mono-modal log normal aerosol size distribution, which has been shown to be a suitable representation of the size distribution of airborne volcanic ash (Wohletz et al., 1989). The distribution is characterized by a spread of $2$ (Wen and Rose, 1994b; Yu et al., 2002; Rybin et al., 2011; Pavolonis et al., 2013b) and the mode radius is translated to obtain different effective radii. The refractive index used in this paper is from measurements of ash from the Eyjafjallajökull eruption (Peters). These properties are calculated every $5\,cm^{-1}$ in the spectral range used by the retrieval, across a range of effective radii from $0.01$–$20\,\mu m$, to create the input for DISORT. Ignoring multiple reflections ...'*

- Page 8, line 1: Please mention the wavenumber (wavelength) of the optical depths.

  *This has been clarified to be at 550nm.*

- Page 9, line 4: Please mention which longitudes are included in the "local" covariance matrix.

  *This has been added to the text. p.9 l.13 'All examples consider a 'local'*

*error covariance matrix, $S_\epsilon$, which is computed using spectra located at all longitudes within the latitude band, $30°–60°$ N, above the Icelandic plume region, ...'*

- Page 9, line 11-13: I do not understand how this explains the decrease in temperature uncertainty. Please clarify. You may also possibly use the simplified model by Prata and Grant (2001) to explain the observed behaviour, see their Eqs. (2)-(5).

  *A reference to Prata and Grant (2001) has been added to the text. p.9 l.24 'It is also known that discerning ash plumes from meteorological cloud is challenging when the temperature contrast with the surface is very small (Prata and Grant, 2001).'*

- Page 10, line 4: Please specify the threshold value.

  *This has been added to the text. p.11 l.13 '...but also only consider the retrieval a success if it converges within 10 iterations and the normalised cost is below a specified threshold (default is 2)...'*

- Page 10, line 8: Please mention what the average retrieved surface temperature including standard deviation. How does it compare to ECMWF values for the area?

  *As mentioned in response to the referees first comment, the authors do not believe that the quantities are directly comparable.*

- Page 11-12, lines 2-8: Do the MODIS and IASI retrievals use the same ash type and size distributions? If yes, please state so. If not, please state how any differences affect the comparison results.

  *Yes the retrievals do both use the same ash type and assumed distribution. This has been clarified in the text p13 l.22 'These properties are the same as those assumed in the IASI retrieval.'*

• Page 12, line 8: What are the units of the number 2.6?

*The units have been confirmed as g cm$^{-3}$*

• Page 12, line 12: Several MODIS pixels cover one IASI pixel. Please mention how the MODIS ash optical propertis vary across the IASI pixels. This variability may be included as vertical error bars in Fig. 6.

*The variability of the MODIS data across the IASI pixel, given as the standard deviation of the averaged values, has been added as error bars to Fig. 6. Please note that, due to updates in the algorithm that reduced the cost of some retrievals, there are additional points to the plot previously presented. The IASI retrieval error has also been added for completeness. Changes in the text: Fig 6. caption: 'Comparison of AOD at $11\,\mu m$ retrieved from IASI and MODIS during the Eyjafjallajköull eruption. The error bars show the associated IASI retrieval error and the standard deviation of the MODIS retrievals that were aggregated across the IASI pixel.'*

• Page 12, line 19: Please mention what the "imposed quality controls" are.

*The values have been included at p.14 l.12 'These measures ensure the output is sensible and realistic (e.g. the plume top altitude is not below the surface or the effective radius negative) and the normalised cost funtion must be below an imposed threshold of 2 for IASI and 5 for MODIS.'*

• Page 12, line 27: Numbers for the "goodness" of the correlation may be obtained if fitting a straight line to the data.

*These values have been added. As above, note that the values have altered very slightly due to the new dataset. In text: p.14 l.17 'Reasonable correlation, $r = 0.47$, is observed between the two instruments with an RMSE value of $0.66$. It is visually clear that there is a grouping of pixels where MODIS overestimates the value of AOD compared to IASI. These coincide with the*

*higher MODIS cost values and largest pixel variability (shown as error bars in Fig. 6) at AOD $> 1$. Removing these pixels, the correlation is much improved to $r = 0.64$. This is especially true for lower values of AOD, where the RMSE reduces to $0.2$ (for AOD $< 1$) and $0.17$ (for AOD $< 0.5$). As the AOD increases, the spread of the data also increases with the tendency for MODIS to see a higher AOD than IASI. However, there is a time difference between the data points and therefore, the instruments may not be viewing the same part of the plume, despite attempts to minimise this. Hence, perfect agreement is not expected and the correlation seen is extremely encouraging.'*

- Page 12, line 30: Eyjafjallajköull → Eyjafjallajökull.

  ***This has been corrected in the text.***

- Page 14, lines 17-18: You state that "The retrieved effective size distribution from IASI measurements is consistent with the values from the aircraft measurements, although slightly smaller." Here you state that you retrieve the effective size distribution from IASI measurements. Is this really so? Is it not the effective radius you retrieve based on an assumed size distribution? Please clarify.

  ***The authors agree that this statement is misleading and it has been reworded to be more clear. p.16 l.13 ' The distribution of retrieved effective radius from IASI measurements is consistent with the values from the aircraft measurements, although slightly smaller. '***

- Page 14, lines 15-20: When comparing effective radii, please provide numbers for the IASI effective radius. This you may obtain by fitting a curve to the histogram in Fig. 7 and thus obtain an estimate of the IASI effective radius.

  ***Fig. 7 is a histogram of retrieved effective radius. It is meant to demonstrate the variation of effective radius within the ash plume. If the referee would***

*like some representation of the effective radius of the entire plume then the mode of this distribution is a better statistic then forming the effective radius from a distributions of effective radii. This has been added to the table of values.*

- Page 16, lines 23: What is implied by "The colocation for this scene is good"? Please quantify time and spatial differences.

    *The information has been added: 'The colocation for this scene is good (within $1.5$ hrs and $50$ km), with the CALIOP track directly crossing the retrieved IASI plume at latitudes above $55°$ N.'*

- Page 16, line 25: 10 → Fig. 10.

    *This has been corrected in the text.*

- Page 16, line 26: Please quantify "good agreement".

    *Further statistics have been added to the text. p.19 l.3 '...The outliers are reflected in the RMSE value for the height comparison, which is 2.5 km ($r = 0.31$). However, upon removing the optically thick outliers from the scene, this reduces the RMSE difference to 0.8 km and increases the correlation to $r = 0.41$. The comparison for another well co-located scene is also shown in Fig. 10 for the $11^{th}$ May 2010, which again shows good agreement with $r = 0.46$ and an RMSE value of $0.9$ km.*

    *Comparisons are not shown for all scenes individually, however, Fig. 11 shows the comparison for all points across all scenes. Some scenes have far fewer co-located pixels but do confirm that there is agreement between the CALIOP and IASI derived altitude range with the values largely occurring between $2$ and $6$ km. Despite good correlation in individual scenes, it is very low for all pixels, $r = 0.12$, with RMSE of $2.1$ km. Visually, it can be seen that there are cases where the retrieval fails to fully capture the higher*

*altitude plumes and there is an underestimation of the plume top height (as previously described), however, this is for only two of the scenes and given the time difference between the satellite overpasses,...'*

- Page 16, line 30: Please beaware that the altitudes in Stohl et al. (2010) are derived from IASI and SEVIRI measurements using an inversion procedure. They only include the altitude of the fine ash that may be dispersed. Thus their use as a reference here is dubious. For the altitude of the plume above the volcanic vent the Arason et al. (2011) reference is maybe more appropriate.

  *The authors take the referees point on board and have changed the reference.*

- Page 17, Fig. 8: What is shown by the solid line in the Figure?

  *It has been made clearer in the caption what each part of the figure illustrates. Fig. 8 Caption: 'An example of the derived CALIOP cloud top heights are shown (as the solid line) for an overpass of Grimsvötn on the $22^{nd}$ May 2011. The CALIOP derived height at locations co-located with IASI pixels are illustrated by triangles and the background shows the backscatter seen by CALIOP.'*

- Page 18, lines 14-15: This could be due to the ash cloud being above an optically thick low altitude cloud, case b in Fig. 2. If the below cloud is optically thick the retrieved surface temperature should represent that of the cloud and not the Earth's surface. Thus it would be interesting to know the retrieved surface temperatureds for these pixels and how they compare with the surface temperatures from for example ECMWF.

  *The authors believe that under a thick cloud there should be little sensitivity to surface temperature and therefore the retrieved surface temperature (or effective radiating temperature) should be close to the a priori (ECMWF).*

*However, please see the authors comments to the first point regarding the 'surface temperature' and comparisons to ECMWF.*

- Page 20, lines 16: You state "skewing towards slightly smaller particles due to viewing a larger area of the plume." However, I can not see that you have given evidence anywhere that the larger area is the reason. Yes, you speculate that this is the reason, but hard facts are needed to be able to firmly state this. Please clarify.

   *The authors accept that this statement is not specific enough and have clarified that it is a potential cause. p.22 l.18 'Aircraft campaigns during the Eyjafjallajökull eruption confirm that the retrieved distribution of effective radii from IASI is in line with the aircraft measurements, skewing towards slightly smaller particles potentially due to viewing a larger area of the plume and therefore a slightly different distribution of the ash.'*

Please also note the supplement to this comment:
http://www.atmos-meas-tech-discuss.net/amt-2016-143/amt-2016-143-AC2-supplement.pdf

**Supplement:**

**Retrieval of ash properties from IASI measurements**

Lucy J. Ventress[1], Roy G. Grainger[2], Gregory McGarragh[3], Elisa Carboni[2], and Andrew J. Smith[1]

[1]National Centre for Earth Observation, Atmospheric, Oceanic and Planetary Physics, University of Oxford, Parks Road, Oxford OX1 3PU, U.K.
[2]COMET, Atmospheric, Oceanic and Planetary Physics, University of Oxford, Parks Road, Oxford, OX1 3PU, U.K.
[3]Atmospheric, Oceanic and Planetary Physics, University of Oxford, Parks Road, Oxford, OX1 3PU, U.K.

*Correspondence to:* L. J. Ventress (lucy.ventress@physics.ox.ac.uk)

**Abstract.** A new optimal estimation algorithm for the retrieval of volcanic ash properties has been developed for use with the Infrared Atmospheric Sounding Interferometer (IASI). The retrieval method uses the wavenumber range $680$–$1200\,\text{cm}^{-1}$, which contains window channels, the $CO_2$ $\nu_2$ band (used for the height retrieval), and the $O_3$ $\nu_3$ band.

Assuming a single infinitely (geometrically) thin ash plume and combining this with the output from the radiative transfer model RTTOV, the retrieval algorithm produces the most probable values for the ash optical depth (AOD), particle effective radius, plume top height and effective radiating temperature. A comprehensive uncertainty budget is obtained for each pixel. Improvements to the algorithm through the use of different measurement error covariance matrices is explored, comparing the results from a sensitivity study of the retrieval process using covariance matrices trained on either clear-sky or cloudy scenes. The result showed that, due to the smaller variance contained within it, the clear-sky covariance matrix is preferable. However, if the retrieval fails to pass the quality control tests, the cloudy covariance matrix is implemented.

The retrieval algorithm is applied to scenes from the Eyjafjallajökull eruption in 2010 and the retrieved parameters are compared to ancillary data sources. The ash optical depth gives a Root Mean Square Error (RMSE) difference of $0.46$ when compared to retrievals from the MODIS instrument for all pixels and an improved RMSE of $0.2$ for low optical depths (AOD$<$ $0.1$). Measurements from the FAAM and DLR flight campaigns are used to verify the retrieved particle effective radius, with the retrieved distribution of sizes for the scene showing excellent consistency. Further, the plume top altitudes are compared to derived cloud-top altitudes from the CALIOP instrument and show agreement with RMSE values of less than $1\,\text{km}$.

**1 Introduction**

The detection of volcanic ash and the retrieval of its properties has become a topic of increasing interest following the eruption of Eyjafjallajökull in 2010. Volcanoes are responsible for the emission of large quantities of aerosol particles and gases, such as $H_2O$, $CO_2$ and $SO_2$, into the atmosphere. The particles created during a volcanic event are classified according to size with the smaller solid particles (radii $< 2\,\text{mm}$) referred to as volcanic ash (Schmid, 1981). These particles can have significant effects upon the Earth's radiation balance, air quality and the aviation industry; the worst outcome in the latter case resulting in engine failure (Grainger et al., 2013; Casadevall, 1994). Through the analysis of spectral information from satellite infrared spectrometers (such as the Atmospheric Infrared Sounder, AIRS, the Tropospheric Emission Spectrometer, TES, and

the Infrared Atmospheric Sounding Interferometer, IASI), the optical and physical properties of volcanic ash can be derived (e.g. the mass of ash contained within the plume) and these can be used to calculate the parameters most useful in ensuring safe air travel (Dubuisson et al., 2014). Several different approaches have been applied to the infrared spectra of different volcanic plumes, including methods based upon optimal estimation (Clarisse et al., 2010; Francis et al., 2012; Pavolonis et al.,

5   2013a), singular value decomposition (Klüser et al., 2013) and split-window (Wen and Rose, 1994a; Prata and Grant, 2001). These methods have been applied to both hyperspectral and broad band satellite instruments, each of which have advantages and disadvantages. For example, IASI, on board MetOp-A, has a wealth of spectral information with over 8000 wavenumber channels allowing the potential to retrieve many parameters. Whereas the Advanced Very-High Resolution Radiometer, AVHRR, a broad band instrument, has only 6 channels to extract information from meaning more assumptions must be made

10  about the state. However, the spatial coverage of AVHRR is much greater than IASI with a footprint of $\sim 1\,\text{km}$ compared to IASI's 12km footprint, giving far more measurements within a volcanic plume.

Presented here is a new optimal estimation algorithm for the retrieval of volcanic ash properties that has been developed for IASI to take advantage of its spectral information, which could be further adapted for use with other hyperspectral satellite instruments. The retrieval method uses the wavenumber range 680–1200 $\text{cm}^{-1}$, which contains window channels, the $CO_2$ $\nu_2$

15  band, and the $O_3$ $\nu_3$ band.

In this work the Oxford-RAL Retrieval of Aerosol and Cloud (ORAC) algorithm (Thomas et al., 2009a; Poulsen et al., 2012), which was successfully applied to the retrieval of volcanic $SO_2$ by Carboni et al. (2012) through the addition of a generalised error covariance matrix, is adapted for use with volcanic ash. The method uses an optimal estimation retrieval algorithm to obtain probable values for the ash optical depth (AOD), particle effective radius, plume top height and effective

20  radiating temperature. The reliability of the retrieved parameters is discussed with a focus upon the validation of the height product, which, in other methods is usually assumed to be some fixed value. Identifying the ash plume top height is a challenge for remote sensing as it is a critical parameter for the initialisation of algorithms that numerically model the evolution and transport of a volcanic plume (Grainger et al., 2013). Validation of the parameters is carried out through comparisons to the derived plume top height from the Cloud- Aerosol Lidar with Orthogonal Polarization (CALIOP), a retrieved AOD from the

25  MODerate-resolution Imaging Spectroradiometer (MODIS) and particle effective radius measurements from the Facility for Airborne Atmospheric Measurements (FAAM) and Deutsches Zentrum f˙r Luft- und Raumfahrt e.V. (DLR) flight campaigns. The examples shown are for the Icelandic volcano Eyjafjallajökull (2010) due to the co-location of satellite data being greatest near the poles.

In this paper the fundamental instrument used in the analysis is described in section 2 followed by the introduction of the

30  retrieval algorithm and forward model in section 3. The sensitivity of the retrieval to different error covariance matrices is discussed in section 4 and, after the results from comparisons of the retrieved IASI parameters with alternative data sources are shown in section 5, conclusions are made in section 6.

**2  IASI**

IASI, on board the MetOp platforms, is a series of three identical Fourier transform spectrometers designed primarily to provide data to be assimilated for use in numerical weather prediction (NWP). The instrument is a Michelson interferometer covering the mid-infrared (IR) from $645$–$2760\,cm^{-1}$ ($3.62$–$15.5\,\mu m$) with a spectral resolution of $0.5\,cm^{-1}$ (apodised) and a pixel diameter at nadir of $12\,km$. MetOp's sun-synchronous polar orbit and IASI's wide swath width means that global coverage is achieved twice daily with the day-time descending node overpass at 09:30 local time for IASI-A (Siméoni et al., 1997; Chalon et al., 2001; Hébert et al., 2004). Since aerosol fields have high spatial and temporal variability, regular views of the same area are essential to characterise plume evolutions. Therefore, IASI's characteristics make it a very useful tool for the observation of larger aerosol particles (such as dust and volcanic ash). The work shown here uses IASI level 1c radiance data obtained from the British Atmospheric Data Centre (BADC) archive.

**3  Retrieval Method**

**3.1  Optimal Estimation Algorithm**

An optimal estimation scheme has been developed to retrieve the properties of volcanic ash plumes. The method analyses the brightness temperature spectra from IASI to retrieve the following parameters: ash optical depth (at a reference wavelength of $550\,nm$), ash effective radius ($\mu m$), ash plume top height (km) and effective radiating temperature (K).

An ash detection method, based upon the trace gas detection method described by Walker et al. (2011) and applied to volcanic ash by Sears et al. (2013), flags IASI pixels for the presence of volcanic ash. The detection procedure looks for departures in a spectrum from an expected background covariance. An ensemble training set of IASI data, assumed to contain no extraordinary ash concentrations, is used to create a generalised error covariance matrix that contains the spectral variability caused by interfering trace species and clouds as well as the IASI instrument noise. A least squares fit retrieval is carried out to retrieve the ash optical depth at three assumed altitudes; $400\,mb$, $600\,mb$ and $800\,mb$. The pixel is flagged if the ash optical depth at any of the altitudes passes a given threshold. In previous work, the presence of volcanic $SO_2$ has been used as a proxy for the location of volcanic ash, therefore, pixels are also flagged for $SO_2$ in the same manner as Carboni et al. (2012) and the retrieval is subsequently calculated for pixels that are flagged to contain either a positive ash or $SO_2$ signal. The ash and $SO_2$ flags are produced in near-real time and the results are publicly available within 3 hours of measurement at http://www.nrt-atmos.cems.rl.ac.uk/.

[revised manuscript text omitted]

30 ash are calculated using Mie theory (Grainger et al. (2004); code available at: http://www.eodg.atm.ox.ac.uk/MIE/index.html) and external mixing. The ash patrticles are assumed to be spherical with a mono-modal log normal aerosol size distribution,

[Figure]

**Figure 3.** Schematic showing the atmospheric interactions simulated in the radiative transfer forward model

which has been shown to be a suitable representation of the size distribution of airborne volcanic ash (Wohletz et al., 1989). The distribution is characterized by a spread of 2 (Wen and Rose, 1994b; Yu et al., 2002; Rybin et al., 2011; Pavolonis et al., 2013b) and the mode radius is translated to obtain different effective radii. The refractive index used in this paper is from measurements of ash from the Eyjafjallajökull eruption (Peters). These properties are calculated every $5\,\mathrm{cm}^{-1}$ in the spectral range used by the retrieval, across a range of effective radii from $0.01\text{–}20\,\mu\mathrm{m}$, to create the input for DISORT.

Ignoring multiple reflections between the layer and the surface, the 'ash' TOA radiance, $R_\bullet^\uparrow$, can be expressed as

$$R_\bullet^\uparrow = R_\mathrm{bl}^\uparrow T_\mathrm{l} T_\mathrm{al} + B_\mathrm{l} \epsilon_\mathrm{l} T_\mathrm{al} + R_\mathrm{al}^\downarrow R_\mathrm{l} T_\mathrm{al} + R_\mathrm{al}^\uparrow, \tag{5}$$

where the terms on the right hand side correspond to, in order, the upwelling radiance below the ash layer transmitted by the layer and atmosphere above it, the emission from the ash layer, the reflected downwelling radiance above the ash layer and the upwelling radiance contribution from the atmosphere above the ash layer.

**4 Error analysis/Sensitivity Study**

An advantage of the optimal estimation framework is it provides a rigorous estimation of the uncertainty in the retrieved state. The *a posteriori* error covariance matrix, $\mathbf{S}_x$, can be written as

$$\mathbf{S}_x = (\mathbf{K}^\mathrm{T} \mathbf{S}_\epsilon^{-1} \mathbf{K} + \mathbf{S}_a^{-1})^{-1}, \tag{6}$$

where $\mathbf{K}$ is the Jacobian, which represents how the measurement spectrum is expected to change given a perturbation to the state. The diagonals of $\mathbf{S}_x$ provide the expected variance on the retrieved state vector elements and, hence, the square root of the diagonals give the uncertainty in each retrieved parameter. The optimal estimation retrieval produces the most

probable values for; ash optical depth, particle effective radius, plume top height and effective radiating temperature, each with associated uncertainties. Further, from these values and an assumed ash density, the ash mass in the plume can be derived (see http://eodg.atm.ox.ac.uk/user/grainger/research/aerosols.pdf).

An uncertainty analysis was performed using synthetic spectra (adding an ash plume to a reference clear atmosphere) to
5  assess the sensitivity of the retrieved parameters to variations in the state. In the simulations the ash optical depth at $550\,nm$ varied between 0.01 and 10 and the plume top altitude lay between $1000\,mb$ and $100\,mb$ ($\sim 0$–$16\,km$). For the examples shown in Fig. 4 the effective radius and effective radiating temperature are held constant at $2\,\mu m$ and $291\,K$ respectively. (Simulations were also carried out varying these values but they are not shown here). The a priori uncertainty estimates used were $\pm 1$ for the logarithm of ash optical depth, $\pm 6\,\mu m$ for effective radius, $\pm 150\,mb$ for altitude and $\pm 20\,K$ for effective radiating temperature.
10  The results show the uncertainties in the retrieved parameters and the Degrees of Freedom for Signal (DFS) within the retrieval for different scenarios. The DFS is a figure of merit that expresses the information contained in a retrieval by compressing the information within the retrieval error covariance matrix into a single scalar quantity. Essentially, it provides the number of independent pieces of information available in an estimate of the state. All examples consider a 'local' error covariance matrix, $S_\epsilon$, which is computed using spectra located at all longitudes within the latitude band, $30°$–$60°$ N, above the Icelandic plume
15  region, which is calculated as described in section 3.2. The resultant uncertainties are shown for optimal estimation retrievals using both the clear and cloudy covariance matrices.

Using the clear covariance gives consistently larger DFS available for all combinations of parameters, with optically thick plumes at lower altitudes having nearly 0.5 DFS more than the results using the cloudy covariance. The impact of this difference in DFS can be seen in the uncertainty in effective radiating temperature, where the more optically thick plumes have
20  a significantly larger uncertainty. Interestingly, and perhaps unexpectedly, the effective radiating temperature uncertainty improves when the ash layer is at the highest altitudes. This is due to the substantially larger amount of atmosphere below a plume at $16\,km$ (as opposed to $8\,km$). This leads to an increase in the fraction of the total radiance across the window regions contributed by the atmosphere below the plume, and conversely, a decrease in the fraction of the total radiance that comes from the emission of the plume itself. It is also known that discerning ash plumes from meterological cloud is challenging
25  when the temperature contrast with the surface is very small (Prata and Grant, 2001). Additionally, the region of low effective radiating temperature uncertainty (high altitude and optically thick plume) coincides with the region with the lowest plume top altitude uncertainty. Essentially, the more accurately we are able to retrieve the height of the plume, the more information that is available to improve the effective radiating temperature estimate. This behaviour has been seen in several other instances, such as Ackerman (1997) and 
[revised manuscript text omitted]

The ash particles are assumed to be spherical with a log-normal size distribution and the size distribution averaged spectral optical properties (extinction coefficient, single scattering albedo and phase function are calculated using Mie theory. Since the width of the distribution is not a retrieval parameter it must be assumed and a standard deviation of 2.0 is the value adopted for the ORAC retrieval. The complex index of refraction must also be assumed for which we use values measured from Eyjafjallajökull ash samples (Peters). These properties are the same as those assumed in the IASI retrieval.

The ash optical properties are further used as input to the plane parallel radiative transfer solver DISORT to compute scalar spectral reflection, transmission and emission operators used in a "fast" forward model, details of which are described in McGarragh et al. (2016), and stored in LUTs as a function of the retrieved 0.55-$\mu$m optical thickness and effective radius, in addition to the solar and satellite geometry. The optical thickness at 11 $\mu$m is obtained directly from the ratio of the extinction coefficient at 11 $\mu$m to that at 0.55 $\mu$m. The ash plume is assumed to be infinitely thin geometrically allowing for a full decoupling of the ash radiative transfer from that of the clear-sky for which the transmittance and emission are computed with RTTOV from meteorological pressure, temperature, humidity and ozone profiles from the ECMWF ERA-Interim reanalysis product (Dee et al., 2011). Molecular (Rayleigh) scattering is computed according to Bates (1984) from the pressure and temperature profiles. Finally, the surface is characterized with a bidirectional reflectance distribution function (BRDF) for both land (Schaaf et al., 2002) and ocean (Sayer et al., 2010). Specific details regarding the sources of uncertainty are discussed in Thomas et al. (2009b), Sayer et al. (2011) and McGarragh et al. (2016).

**5.1.2 Colocating the data**

IASI and MODIS have very different fields of view and, hence, they must be co-located in order for a comparison to be carried out. The number of MODIS retrievals is far greater than that for IASI due to its better spatial resolution along the track. In order to compare the results, the MODIS data is aggregated onto the IASI resolution, i.e. all MODIS pixels within 6 km of the IASI pixel centre are used to formulate the average. Co-location is assumed if the IASI measurements and the MODIS measurements lie within 50 km and 1 hour of each other.

**5.1.3 Results**

A comparison of the AOD at $11\,\mu$m retrieved by both the IASI and MODIS algorithms is shown in Fig. 6. Although the retrievals provide their AOD output at 550 nm, these values are obtained by spectral extrapolation and the value of AOD at $11\,\mu$m is more appropriate for comparison as it lies within the actual wavenumber range for both instruments. The data shown is from the Eyjafjallajköull eruption in 2010 and only the retrievals that pass the imposed quality control measures for both algorithms are shown (73 coincidences). These measures ensure the output is sensible and realistic (e.g. the plume top altitude is not below the surface or the effective radius negative) and the normalised cost funtion must be below an imposed threshold of 2 for IASI and 5 for MODIS. A further criteria was imposed upon the MODIS data that all of the data points averaged onto the IASI pixel resolution must be flagged as ash by the MODIS algorithm for the aggregated pixel to be used in the comparison. This is to ensure that we are comparing like with like.

Reasonable correlation, $r = 0.47$, is observed between the two instruments with an RMSE value of 0.66. It is visually clear that there is a grouping of pixels where MODIS overestimates the value of AOD compared to IASI. These coincide with the higher MODIS cost values and largest pixel variability (shown as error bars in Fig. 6) at AOD $> 1$. Removing these pixels, the correlation is much improved to $r = 0.64$. This is especially true for lower values of AOD, where the RMSE reduces to 0.2 (for AOD $< 1$) and 0.17 (for AOD $< 0.5$). As the AOD increases, the spread of the data also increases with the tendency for MODIS to see a higher AOD than IASI. However, there is a time difference between the data points and therefore, the instruments may not be viewing the same part of the plume, despite attempts to minimise this. Hence, perfect agreement is not expected and the correlation seen is extremely encouraging.

**5.2 Effective Radius: Comparison to aircraft measurements**

**5.2.1 Aircraft Description**

Immediately following the Eyjafjallajökull eruption in 2010, it became clear that aircraft measurements (both in-situ and remote sounding) were needed in order to validate the ash dispersion forecasts. Two of the European aircraft deployed were the UK's BAe-146 FAAM aircraft (http://www.faam.ac.uk) and Germany's DLR Falcon aircraft (http://www.dlr.de). These aircraft are described in great detail elsewhere (see Marenco et al. (2011), Turnbull et al. (2012) and Newman et al. (2012) for FAAM aircraft; Schumann et al. (2011) for DLR aircraft) and therefore only a brief description is given here.

[Figure]

**Figure 6.** Comparison of AOD at $11\,\mu$m retrieved from IASI and MODIS during the Eyjafjallajköull eruption. The error bars show the associated IASI retrieval error and the standard deviation of the MODIS retrievals that were aggregated across the IASI pixel.

On board the FAAM aircraft were instruments capable of taking in situ and remotely sounded measurements. The in situ observations come from two wing-mounted optical particle counters: a passive cavity aerosol spectrometer probe for particles with size distributions of diameter $0.1$–$3\,\mu$m and a cloud and aerosol spectrometer for particles of diameter $0.6$–$50\,\mu$m. Essentially these equate to fine and coarse mode aerosol respectively. The principal remotely sounded observations came from the on board lidar instrument; an ALS450 elastic backscatter lidar mounted to view in the nadir, which operates at a wavelength of $355\,$nm and has a footprint ranging from $7$–$11\,$km.

The DLR aircraft used the same instruments as the FAAM aircraft, however, the assumptions made in the calculation of the size distributions were different. Values for the optical properties (refractive index and shape) of the particles must be assumed as the response of the detectors is dependent upon these as well as the size (Turnbull et al., 2012). The DLR results assume spherical particles whereas the FAAM aircraft provide results for both spherical and irregular particles, with an additional result assuming the refractive indices of the DLR model, showing how the differing assumptions affect the results.

**5.2.2 Results**

Turnbull et al. (2012) provide in situ measurements of the Eyjafjallajköull volcanic ash cloud on $17^{\text{th}}$ May 2010 from both the FAAM and DLR flight campaigns. Despite no actual overlap in the flight paths of the aircraft, a worthwhile comparison

is still possible and here we further compare to the IASI retrievals on the same day. Values for the geometric mean diameter and standard deviation of the particle size distribution are given from both aircraft for both the fine and coarse particle modes. In order to compare these results to the retrieved IASI parameters, they must be converted into number weighted mean radius, $r_N$, by

$$r_N = \frac{D_g}{2} e^{-3\sigma^2}, \tag{7}$$

where $D_g$ is the geometric mean diameter by volume and $\sigma$ is the logarithm of the geometric standard deviation, $S$. Further, due to IASI having sensitivity to both the fine and coarse modes, they are combined to calculate the effective radius, $r_{\text{eff}}$, using

$$r_{\text{eff}} = \frac{\sum_i m_i \exp\left[3\log r_{N,i} + \frac{9}{2}\sigma_i^2\right]}{\sum_i m_i \exp\left[2\log r_{N,i} + 2\sigma_i^2\right]}, \tag{8}$$

where the mixing ratios $m_i$ are the relative weight of each mode and a log-normal distribution is assumed. The size distributions obtained from the aircraft measurements can be seen in Table 1. Further, a histogram of the effective radius retrieved by IASI across all scenes containing the volcanic ash plume on the 17[th] May can be seen in Fig. 7.

**Table 1.** The parameters observed by aircraft on the 17[th] May 2010 (geometric mean diameter by volume, geometric standard deviation, and relative weight by mass, taken from Turnbull et al. (2012)) and the effective radius calculated from the size distributions. The mode effective radius measured for the entire ash plume observed by IASI is also shown.

[revised manuscript text omitted]